# DIFFSSR: Stereo Image Super-resolution Using Differential Transformer

**Dafeng Zhang**
Samsung R&D Institute China-Beijing (SRC-B)
dfeng.zhang@samsung.com

## Abstract

In the field of computer vision, the task of stereo image super-resolution (StereoSR) has garnered significant attention due to its potential applications in augmented reality, virtual reality, and autonomous driving. Traditional Transformer-based models, while powerful, often suffer from attention noise, leading to suboptimal reconstruction issues in super-resolved images. This paper introduces DIFFSSR, a novel neural network architecture designed to address these challenges. We introduce the Diff Cross Attention Block (DCAB) and the Sliding Stereo Cross-Attention Module (SSCAM) to enhance feature integration and mitigate the impact of attention noise. The DCAB differentiates between relevant and irrelevant context, amplifying attention to important features and canceling out noise. The SSCAM, with its sliding window mechanism and disparity-based attention, adapts to local variations in stereo images, preserving details, and addressing the performance degradation due to misalignment of horizontal epipolar lines in stereo images. Extensive experiments on benchmark datasets demonstrate that DIFF-SSR outperforms state-of-the-art methods, including NAFSSR and SwinFIRSSR, in terms of both quantitative metrics and visual quality. Code is available at https://github.com/Zdafeng/DIFFSSR.

## 1 Introduction

The rapid advancements in deep learning have led to significant progress in the field of single image super-resolution (SISR) [1–4], where the goal is to reconstruct high-quality images from their degraded low-resolution counterparts. The impressive results achieved by deep learning models in this domain have naturally piqued the interest of researchers in leveraging the complementary information present in stereo image pairs to further enhance resolution through deep learning methods [5, 6, 4]. This has directed the focus towards the exploration of new fundamental neural network architectures and utilization of the complementary information present in the left and right views for stereo image super-resolution (StereoSR).

Among the various neural network architectures that have emerged, SwinIR [3] stands out as a simple yet effective baseline for image super-resolution task. It introduces the Swin Transformer [7] as a powerful backbone to capture long-range dependencies and global interactions within images, and achieves state-of-the-art (SOTA) performance in various restoration tasks, including image super-resolution, denoising, and JPEG compression artifact reduction. SwinFIR [4], an extension of SwinIR, incorporates Fast Fourier Convolution (FFC) [8] to further enhance the model's ability to capture global information and achieves a new SOTA results in image super-resolution benchmarks. In the field of StereoSR, SwinFIRSSR [4] extends SwinFIR's capabilities by fusing left/right viewpoint features using a stereo cross-attention module (SCAM) [6]. This adaptation allows SwinFIRSSR to effectively utilize the complementary information from stereo pairs, resulting in sharper and more detailed stereo super-resolved images.

39th Conference on Neural Information Processing Systems (NeurIPS 2025).

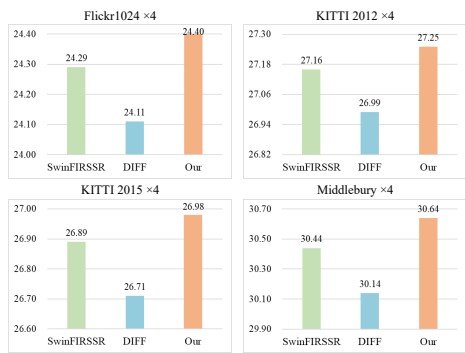

Figure 1: The comparison results with Swin-FIRSSR. DIFF denotes replacing its Transformer with the DIFF Transformer.

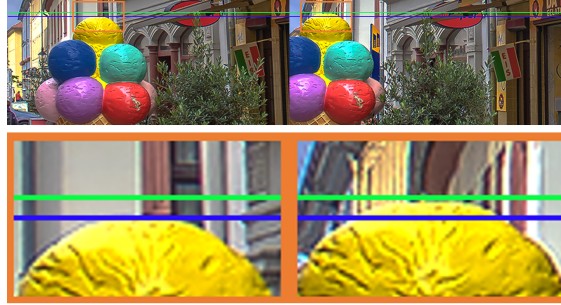

Figure 2: On the horizontal epipolar line, the left and right views are not strictly aligned. Therefore, the performance of SCAM will be hindered.

The core of the Transformer's success lies in its attention mechanism [9], which enables the model to weigh the importance of different tokens in a sequence. However, recent studies have highlighted the challenges faced by large language models (LLMs) in accurately retrieving key information from context, a limitation that stems from the non-negligible attention scores assigned to irrelevant context [10, 11], often termed as attention noise. Therefore, SwinFIRSSR inevitably contains the flaws of the Transformer. Ye *et al.* [11] introduce the Differential Transformer (DIFF Transformer), a novel architecture that amplifies attention to relevant context while effectively canceling out noise. So, can the DIFF Transformer also yield significant performance improvements in visual tasks? We substituted the Transformer architecture in SwinFIRSSR with the DIFF Transformer. However, experimental results revealed that the DIFF Transformer not only failed to enhance performance but also degraded the performance of existing methods, as shown in Figure 1. Consequently, this paper presents a redesign of the DIFF Transformer, tailored for stereo super-resolution task. This is the inaugural exploration of the DIFF Transformer in visual tasks.

In addition to the exploration of the basic network structure, within the domain of StereoSR, several methods also have been developed to leverage the cross-view information present in stereo image pairs. PASSRNet [12] introduced a parallax attention mechanism to effectively handle stereo images with large disparities. iPASSR [5] further improved upon this by incorporating a bidirectional parallax attention module (biPAM) and an inline occlusion handling scheme. NAFSSR [6] employs a Stereo Cross-Attention Module (SCAM) to integrate cross-view information. SCAM is designed to seamlessly blend the simplicity and effectiveness of NAFNet [13] with the distinctive attributes of stereo super-resolution task. By focusing on cross-view attention, SCAM enhances the model's ability to integrate features from both views, improving the overall quality of super-resolved images. While SCAM is effective for cross-view features fusion, it only focuses on features corresponding to the horizontal epipolar line. If image distortion or stereo correction algorithm errors cause local misalignment in stereo images, it will hinder the performance of SCAM, as shown in Figure 2.

In this paper, we propose a novel neural network structure for StereoSR tasks based on the DIFF Transformer, named DIFFSSR. We have re-engineered the DIFF Transformer and introduced the Diff Cross Attention Block (DCAB) to address the inapplicability of the existing DIFF Transformer in visual tasks. Additionally, our DCAB also effectively increases the opportunity for information exchange between the two stereo images, alleviating the issue of insufficient communication between the left and right view information in SwinFIRSSR, which can further enhance the performance of super-resolution. We have also redesigned the Stereo Cross-Attention Module (SCAM) and introduced the Sliding SCAM (SSCAM), which addresses the performance degradation of SCAM due to the misalignment of horizontal epipolar lines in stereo images. Through the aforementioned improvements, our proposed DIFFSSR has achieved state-of-the-art (SOTA) performance across multiple benchmark datasets and significantly outperformed SwinFIRSSR.

The contributions can be summarized as follows:

- We introduced a novel neural network architecture named DIFFSSR, specifically designed for stereo image super-resolution tasks. This architecture is tailored to leverage the complementary information present in stereo image pairs to enhance resolution.

- We proposed the Diff Cross Attention Block (DCAB), a module engineered to address the inapplicability of the existing DIFF Transformer in visual tasks. The DCAB is designed to differentiate between relevant and irrelevant context, effectively amplifying attention to important features and canceling out noise.

- We proposed the Sliding Stereo Cross-Attention Module (SSCAM), an innovative redesign of the traditional Stereo Cross-Attention Module (SCAM). The SSCAM addresses the performance degradation due to misalignment of horizontal epipolar lines in stereo images by employing a sliding window mechanism.

- Through the integration of the DCAB and SSCAM within the DIFFSSR framework, we achieved state-of-the-art (SOTA) performance across multiple benchmark datasets. This was demonstrated through both quantitative metrics and qualitative visual comparisons, outperforming existing methods and validating the effectiveness of our proposed architecture in enhancing stereo image super-resolution.

## 2 Related Work

The pursuit of enhancing image resolution has been a longstanding endeavor in the field of computer vision. Particularly, Stereo Image Super-Resolution (StereoSR) has garnered significant attention due to its potential applications in augmented reality, virtual reality, and autonomous driving.

### 2.1 Single Image Super-Resolution (SISR)

SISR has witnessed remarkable progress with the advent of deep learning. Early methods like SRCNN [1] introduced the concept of using convolutional neural networks for super-resolution tasks. Subsequent works, such as VDSR [14] and EDSR [15], improved upon these foundations by employing deeper network architectures and skip connections to mitigate gradient degradation issues. RCAN [2] further advanced SISR by incorporating a recursive residual structure and attention mechanisms [16], which allowed the model to capture contextual information effectively. These methods, while effective, primarily operate on convolutional neural network, limiting their ability to encode rich contextual information. Based on the Swin Transformer, SwinIR [3] achieves state-of-the-art performance in SISR by leveraging the Transformer's ability to capture long-range dependencies and global information effectively. As an extension of SwinIR, SwinFIR [4] incorporates fast Fourier convolution to enhance the model's ability to capture global information, resulting in improved super-resolution performance.

### 2.2 Stereo Image Super-Resolution (StereoSR)

StereoSR leverages the complementary information from stereo image pairs to enhance resolution. Early approaches, such as StereoSR [17], utilized disparity priors to improve the quality of super-resolved images. PASSRNet [12] introduced the Parallax Attention Mechanism Module (PAM) to address the challenge of varying disparities in stereo images. iPASSR [5] incorporates a Bidirectional Parallax Attention Module (biPAM) and an inline occlusion handling scheme. The biPAM is designed to symmetrically process both left-to-right and right-to-left parallax information, leveraging the symmetry cues in stereo images for enhanced super-resolution. NAFSSR [6] stands out as a lightweight approach that integrates NAFNet's simplicity with the distinctive attributes of StereoSR tasks through the use of stereo cross-attention modules (SCAM). Recently, Transformer-based architectures have been explored in StereoSR due to their ability to capture long-range dependencies. SIR-Former [18] pioneered the use of transformers in StereoSR, employing a cross-attention module to learn epipolar line relationships and a transformer-based fusion module for accurate cross-view feature integration. Steformer [19] leveraged self-attention to capture both cross-view and intra-view information in stereo images, ensuring reliable stereo correspondence and effective cross-view integration. Swin-FIRSSR [4] utilizes the strengths of the Swin Transformer and fast Fourier convolution to achieve SOTA performance in StereoSR. Specifically, it integrates frequency domain knowledge through fast Fourier convolution, enhancing the model's ability to capture global information. However, similar to other Transformer-based models, SwinFIRSSR faces challenge such as increasing attention noise.

# 3 Methodology

## 3.1 Preliminary: Differential Transformer

The Differential Transformer [11] was originally proposed to mitigate the problem of attention noise, a phenomenon where conventional attention mechanisms assign non-negligible weights to irrelevant tokens, thereby diluting the representation of key information. To address this limitation, Ye *et al.* [11] proposed a differential attention mechanism that constructs two complementary attention branches: one emphasizing relevant information and the other modeling irrelevant or noisy signals. The contrast between the two branches effectively suppresses shared distractions while amplifying meaningful information.

Given an input feature sequence $x \in \mathbb{R}^{N \times C}$, the two sets of query and key projections are generated as:

$$[Q_1, Q_2] = xW_Q, \quad [K_1, K_2] = xW_K, \quad V = xW_V, \tag{1}$$

where $W_Q, W_K, W_V \in \mathbb{R}^{C \times C}$ denote the learnable projection matrices. $Q_1, Q_2, K_1, K_2 \in \mathbb{R}^{N \times \frac{C}{2}}$ and $V \in \mathbb{R}^{N \times C}$ denote the query, key and value. $N$ and $C$ are the number of tokens and channels, respectively. The differential attention is formulated as:

$$\text{Att}_{diff}(Q, K, V) = \left( S\left( \frac{Q_1 K_1^T}{\sqrt{C}} \right) - \lambda S\left( \frac{Q_2 K_2^T}{\sqrt{C}} \right) \right) V, \tag{2}$$

where $S(\cdot)$ denotes the softmax function and $\lambda$ is a learnable scalar controlling the suppressive branch. The first term enhances attention toward relevant features, while the second term cancels noise-induced activations, yielding cleaner and more discriminative feature maps. $\lambda$ is formulated as:

$$\lambda = exp(sum(\lambda_{q1} * \lambda_{k1})) - exp(sum(\lambda_{q2} * \lambda_{k2})) + \lambda_{init}, \tag{3}$$

where $\lambda_{q1}, \lambda_{q2}, \lambda_{k1}, \lambda_{k2} \in \mathbb{R}^{\frac{C}{2}}$ are learnable parameters, and $\lambda_{init} = 0.8$ in this paper.

This differential attention mechanism enables the model to emphasize meaningful context while reducing redundancy. However, directly applying this mechanism to dense visual data introduces computational efficiency and performance challenges. To overcome these issues, we reformulate and extend the differential attention paradigm for the visual domain, which serves as the conceptual foundation of our proposed architecture, **DIFFSSR**.

The DIFFSSR architecture, as depicted in Figure 3(a), is composed of multiple Diff Cross Attention Blocks (DCAB) and Sliding Stereo Cross-Attention Module (SSCAM). The DCABs are responsible for differentiating between relevant and irrelevant context, amplifying attention to important features and canceling out noise. The SSCAM, on the other hand, with its sliding window mechanism and disparity-based attention, adapts to local variations in stereo images, preserving details, and addressing the performance degradation due to misalignment of horizontal epipolar lines in stereo images. The detail information of DCAB and SSCAM are described in Section 3.3 and Section 3.4.

## 3.2 Overall Architecture

As illustrated in Figure 3(a), the proposed DIFFSSR architecture consists of three main components: (1) a shallow feature extraction module, (2) a deep feature extraction module composed of stacked Diff Cross-Attention Blocks (DCABs), and (3) a reconstruction network.

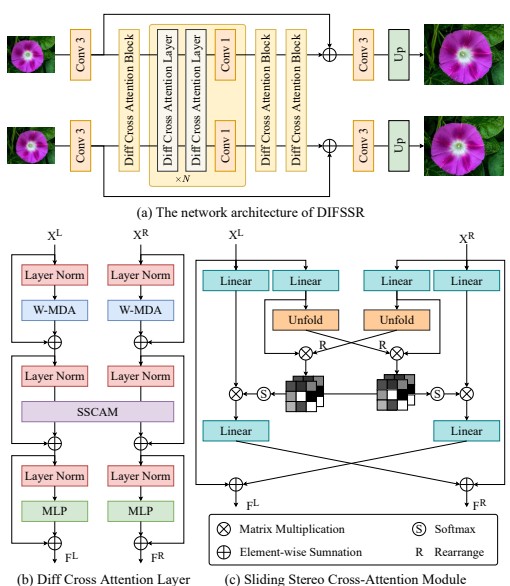

Figure 3: The overall network architecture of DIFFSSR. W-MDA is window-based multi-head differential attention module. SSCAM is sliding stereo cross-attention module.

Given a pair of low-resolution stereo images $X^L, X^R \in \mathbb{R}^{H \times W \times 3}$, the shallow features are first extracted as:

$$F_S^L = H_S(X^L), \quad F_S^R = H_S(X^R), \tag{4}$$

where $H_S(\cdot)$ denotes a $3 \times 3$ convolutional layer used for low-level feature extraction.

The deep feature extraction network is composed of $l$ DCABs, which progressively refine both intra-view and cross-view features:

$$F_D^L = H_D(F_S^L), \quad F_D^R = H_D(F_S^R), \tag{5}$$

where $H_D(\cdot)$ represents the stacked DCABs. Finally, the reconstruction module recovers high-resolution images by combining shallow and deep features through residual learning:

$$I_{SR}^L = H_{REC}(F_D^L + F_S^L), \quad I_{SR}^R = H_{REC}(F_D^R + F_S^R), \tag{6}$$

where $H_{REC}(\cdot)$ consists of a convolutional layer followed by a pixel-shuffle upsampling [20]. This residual design effectively preserves high-frequency textures, focusing the network on reconstructing fine details.

### 3.3 Diff Cross-Attention Block (DCAB)

To extend the Differential Transformer for visual tasks, we propose the Diff Cross-Attention Block (DCAB), which introduces a Window-based Multi-head Differential Attention (W-MDA) and a lightweight stereo feature coupling mechanism. Unlike the original global attention used in text processing, W-MDA operates within local non-overlapping windows, significantly reducing computational complexity while preserving spatial locality. Each DCAB consists of three Diff Cross-Attention Layers (DCAL) followed by a $1 \times 1$ convolution to enhance translation equivariance. Formally, for the $j$-th DCAL in the $i$-th DCAB, the features are updated as:

$$F_{i,j}^L = H_{DCAL}^{(j)}(F_{i,j-1}^L), \quad F_{i,j}^R = H_{DCAL}^{(j)}(F_{i,j-1}^R), \tag{7}$$

and the output of the $i$-th DCAB is obtained as:

$$F_{i,out}^L = H_{conv}^{(i)}(F_{i,N}^L), \quad F_{i,out}^R = H_{conv}^{(i)}(F_{i,N}^R), \tag{8}$$

where $H_{conv}^{(i)}(\cdot)$ denotes a $1 \times 1$ convolutional layer.

Each DCAL integrates three key components: a differential attention module, a sliding stereo cross-attention module (SSCAM), and a feed-forward network (MLP). To adapt the differential attention for visual data, we make two crucial modifications to the original formulation: a. Instead of using the Rotary Position Embedding (RoPE) [21], we adopt *Relative Positional Encoding (RPE)* [7]. RPE explicitly models the relative spatial offsets between pixels, which better preserves spatial consistency within local windows, especially important for stereo correspondence. This substitution improves spatial awareness without introducing additional learnable parameters. b. The original Differential Transformer employs SwiGLU [22, 23] activation within its feed-forward block. However, we replace it with a conventional *two-layer MLP* for two reasons: (1) MLPs require fewer computational resources and are easier to optimize for high-resolution image data; (2) SwiGLU's gating mechanism may impede gradient propagation in dense spatial domains, causing information bottlenecks. Empirically, we find that this replacement leads to better convergence stability and performance. The complete computation process inside one DCAB is summarized as:

$$F_{diff}^L = W\text{-}MDA(LN(F^L)) + F^L, \tag{9}$$

$$F_{diff}^R = W\text{-}MDA(LN(F^R)) + F^R, \tag{10}$$

$$F_{ln}^L = LN(F_{diff}^L), \quad F_{ln}^R = LN(F_{diff}^R), \tag{11}$$

$$F_{sscam}^L, F_{sscam}^R = SSCAM(F_{ln}^L, F_{ln}^R), \tag{12}$$

$$F_{mlp}^L = MLP(LN(F_{sscam}^L)) + F_{sscam}^L, \tag{13}$$

$$F_{mlp}^R = MLP(LN(F_{sscam}^R)) + F_{sscam}^R, \tag{14}$$

where $W\text{-}MDA(\cdot)$, $SSCAM(\cdot)$ and $MLP(\cdot)$ denote the differential attention operator, Sliding Stereo Cross-Attention Module (SSCAM) and MLP, respectively. $LN(\cdot)$ is the layer normalization. $F_{diff}^*$, $F_{sscam}^*$ and $F_{mlp}^*$ denote the outputs of differential attention, SSCAM and MLP, respectively. This structure amplifies critical spatial cues, reduces attention noise, and enables effective communication between left and right stereo features.

### 3.4 Sliding Stereo Cross-Attention Module (SSCAM)

While the DCAB focuses on refining intra-view representations, the Sliding Stereo Cross-Attention Module (SSCAM) is designed to facilitate adaptive cross-view feature fusion, as shown in Figure 3(c). Traditional stereo cross-attention [6] relies on strict horizontal epipolar alignment, which can be disrupted by imperfect calibration or local distortion. To mitigate this issue, SSCAM introduces a sliding-window cross-attention mechanism that dynamically attends to locally relevant disparity regions. The SSCAM functional expression can be described as the Equation (12). In detail, given the left and right features $F^L$ and $F^R$, the fused feature $F^{L \to R}$ are obtained by:

$$Q^L = Unfold(W^{Q^L} F^L), \tag{15}$$

$$K^R = W^{K^R} F^R, \quad V^R = W^{V^R} F^R, \tag{16}$$

$$F^{L \to R} = W^{P^R} Attention(Q^L, K^R, V^R) + F^R, \tag{17}$$

where $W^{Q^L}$, $W^{K^R}$, $W^{V^R}$ and $W^{P^R}$ are the $1 \times 1$ point-wise convolution for left to right attention. $Unfold(\cdot)$ denotes the Unfold operation is used in this paper to implement a sliding window. $Q^L$, $K^R$, and $V^R$ denote the query, key and value for left to right attention. $Attention(\cdot)$ is the Stereo Cross-Attention that can be represented as:

$$Attention(Q, K, V) = S(\frac{QK^T}{\sqrt{C}})V, \tag{18}$$

The fused feature $F^{R \to L}$ are obtained by:

$$Q^R = Unfold(W^{Q^R} F^R), \tag{19}$$

$$K^L = W^{K^L} F^L, \quad V^L = W^{V^L} F^L, \tag{20}$$

$$F^{R \to L} = W^{P^L} Attention(Q^R, K^L, V^L) + F^L, \tag{21}$$

where $W^{Q^R}$, $W^{K^L}$, $W^{V^L}$ and $W^{P^L}$ are the $1 \times 1$ point-wise convolution for right to left attention. $Q^R, K^L, V^L$ denote the query, key and value for right to left attention. Through this formulation, SSCAM achieves robust cross-view correspondence and effectively handles stereo pairs with misaligned epipolar lines, thereby improving reconstruction fidelity.

## 4 Experiments

### 4.1 Implementation Details

**Dataset.** The training dataset for our proposed model consists of a combination of images from the Flickr1024 dataset [12] and the Middlebury dataset [24]. Specifically, we utilize 800 stereo image pairs from Flickr1024 and 60 pairs from Middlebury. Then, low-resolution (LR) images are created by applying bicubic downsampling to the HR images with scaling factors of ×2 and ×4. The resulting LR images are cropped into 32×96 patches with a stride of 16, and their HR counterparts undergo corresponding cropping. For testing, we employ a popular benchmark comprising 20 pairs of images from the KITTI 2012 dataset [25], 20 pairs of images from the KITTI 2015 dataset [26], 112 pairs of images from the Flickr1024 dataset [12], and 5 pairs of images from the Middlebury dataset [24].

**Model Setting.** Our DIFFSSR architecture is primarily composed of a shallow feature extraction network, a deep feature extraction network, and a reconstruction module. The deep feature extraction network is the core component of our model, designed to recover the missing texture details in low-resolution images. It is primarily constructed from 13 Diff Cross Attention Blocks (DCAB), each of which consists of 3 Diff Cross Attention Layers followed by a convolutional operation, with a channel number of 180.

**Training Settings.** The training process for DIFFSSR is conducted over 500,000 iterations with a batch size of 8. We initialize the learning rate at 2e-4 and employ a cosine annealing strategy [27] to gradually decrease the learning rate to 1e-7. Data augmentation techniques, including random horizontal and vertical flips and channel shuffle [4], are applied to enhance dataset diversity. Additionally,

Table 1: Quantitative results achieved by different methods on the KITTI 2012 [25], KITTI 2015 [26], Middlebury [24], and Flickr1024 [12] datasets on the RGB space for **stereo image SR**. $\#P$ represents the number of parameters of the networks. Here, PSNR/SSIM values achieved on both the left images (i.e., *Left*) and a pair of stereo images (i.e., $(Left + Right)/2$) are reported.

| Method | Scale | #P | Left | | | $(Left+Right)/2$ | | | |
|---|---|---|---|---|---|---|---|---|---|
| | | | KITTI 2012 | KITTI 2015 | Middlebury | KITTI 2012 | KITTI 2015 | Middlebury | Flickr1024 |
| EDSR[15] | ×2 | 38.6M | 30.83/0.9199 | 29.94/0.9231 | 34.84/0.9489 | 30.96/0.9228 | 30.73/0.9335 | 34.95/0.9492 | 28.66/0.9087 |
| RDN[29] | ×2 | 22.0M | 30.81/0.9197 | 29.91/0.9224 | 34.85/0.9488 | 30.94/0.9227 | 30.70/0.9330 | 34.94/0.9491 | 28.64/0.9084 |
| RCAN[2] | ×2 | 15.3M | 30.88/0.9202 | 29.97/0.9231 | 34.80/0.9482 | 31.02/0.9232 | 30.77/0.9336 | 34.90/0.9486 | 28.63/0.9082 |
| StereoSR[17] | ×2 | 1.08M | 29.42/0.9040 | 28.53/0.9038 | 33.15/0.9343 | 29.51/0.9073 | 29.33/0.9168 | 33.23/0.9348 | 25.96/0.8599 |
| PASSRnet[12] | ×2 | 1.37M | 30.68/0.9159 | 29.81/0.9191 | 34.13/0.9421 | 30.81/0.9190 | 30.60/0.9300 | 34.23/0.9422 | 28.38/0.9038 |
| IMSSRnet[30] | ×2 | 6.84M | 30.90/- | 29.97/- | 34.66/- | 30.92/- | 30.66/- | 34.67/- | -/- |
| iPASSR[5] | ×2 | 1.37M | 30.97/0.9210 | 30.01/0.9234 | 34.41/0.9454 | 31.11/0.9240 | 30.81/0.9340 | 34.51/0.9454 | 28.60/0.9097 |
| SSRDE-FNet[31] | ×2 | 2.10M | 31.08/0.9224 | 30.10/0.9245 | 35.02/0.9508 | 31.23/0.9254 | 30.90/0.9352 | 35.09/0.9511 | 28.85/0.9132 |
| NAFSSR-T[6] | ×2 | 0.45M | 31.12/0.9224 | 30.19/0.9253 | 34.93/0.9495 | 31.26/0.9254 | 30.99/0.9355 | 35.01/0.9495 | 28.94/0.9128 |
| NAFSSR-S[6] | ×2 | 1.54M | 31.23/0.9236 | 30.28/0.9266 | 35.23/0.9515 | 31.38/0.9266 | 31.08/0.9367 | 35.30/0.9514 | 29.19/0.9160 |
| NAFSSR-B[6] | ×2 | 6.77M | 31.40/0.9254 | 30.42/0.9282 | 35.62/0.9545 | 31.55/0.9283 | 31.22/0.9380 | 35.68/0.9544 | 29.54/0.9204 |
| NAFSSR-L[6] | ×2 | 23.79M | 31.45/0.9261 | 30.46/0.9289 | 35.83/0.9559 | 31.60/0.9291 | 31.25/0.9386 | 35.88/0.9557 | 29.68/0.9221 |
| SwinFIRSSR[4] | ×2 | 23.94M | 31.65/0.9293 | 30.66/0.9321 | 36.48/0.9601 | 31.79/0.9321 | 31.45/0.9413 | 36.52/0.9598 | 30.14/0.9286 |
| MSSFNet[32] | ×2 | 1.80M | 31.37/0.9262 | 30.37/0.9287 | 35.77/0.9555 | 31.53/0.9292 | 31.16/0.9384 | 35.82/0.9553 | 29.45/0.9212 |
| **DIFFSSR-T (Ours)** | ×2 | 1.54M | 31.40/0.9267 | 30.41/0.9295 | 35.89/0.9568 | 31.55/0.9295 | 31.20/0.9391 | 35.94/0.9567 | 29.63/0.9233 |
| **DIFFSSR (Ours)** | ×2 | 19.86M | **31.69/0.9298** | **30.68/0.9327** | **36.61/0.9609** | **31.84/0.9327** | **31.47/0.9418** | **36.65/0.9607** | **30.27/0.9301** |
| EDSR[15] | ×4 | 38.9M | 26.26/0.7954 | 25.38/0.7811 | 29.15/0.8383 | 26.35/0.8015 | 26.04/0.8039 | 29.23/0.8397 | 23.46/0.7285 |
| RDN[29] | ×4 | 22.0M | 26.23/0.7952 | 25.37/0.7813 | 29.15/0.8387 | 26.32/0.8014 | 26.04/0.8043 | 29.27/0.8404 | 23.47/0.7295 |
| RCAN[2] | ×4 | 15.4M | 26.36/0.7968 | 25.53/0.7836 | 29.20/0.8381 | 26.44/0.8029 | 26.22/0.8068 | 29.30/0.8397 | 23.48/0.7286 |
| StereoSR[17] | ×4 | 1.42M | 24.49/0.7502 | 23.67/0.7273 | 27.70/0.8036 | 24.53/0.7555 | 24.21/0.7511 | 27.64/0.8022 | 21.70/0.6460 |
| PASSRnet[12] | ×4 | 1.42M | 26.26/0.7919 | 25.41/0.7772 | 28.61/0.8232 | 26.34/0.7981 | 26.08/0.8002 | 28.72/0.8236 | 23.31/0.7195 |
| SRRes+SAM[33] | ×4 | 1.73M | 26.35/0.7957 | 25.55/0.7825 | 28.76/0.8287 | 26.44/0.8018 | 26.22/0.8054 | 28.83/0.8290 | 23.27/0.7233 |
| IMSSRnet[30] | ×4 | 6.89M | 26.44/- | 25.59/- | 29.02/- | 26.43/- | 26.20/- | 29.02/- | -/- |
| iPASSR[5] | ×4 | 1.42M | 26.47/0.7993 | 25.61/0.7850 | 29.07/0.8363 | 26.56/0.8053 | 26.32/0.8084 | 29.16/0.8367 | 23.44/0.7287 |
| SSRDE-FNet[31] | ×4 | 2.24M | 26.61/0.8028 | 25.74/0.7884 | 29.29/0.8407 | 26.70/0.8082 | 26.43/0.8118 | 29.38/0.8411 | 23.59/0.7352 |
| NAFSSR-T[6] | ×4 | 0.46M | 26.69/0.8045 | 25.90/0.7930 | 29.22/0.8403 | 26.79/0.8105 | 26.62/0.8159 | 29.32/0.8409 | 23.69/0.7384 |
| NAFSSR-S[6] | ×4 | 1.56M | 26.84/0.8086 | 26.03/0.7978 | 29.62/0.8482 | 26.93/0.8145 | 26.76/0.8203 | 29.72/0.8490 | 23.88/0.7468 |
| NAFSSR-B[6] | ×4 | 6.80M | 26.99/0.8121 | 26.17/0.8020 | 29.94/0.8561 | 27.08/0.8181 | 26.91/0.8245 | 30.04/0.8568 | 24.07/0.7551 |
| NAFSSR-L[6] | ×4 | 23.83M | 27.04/0.8135 | 26.22/0.8034 | 30.11/0.8601 | 27.12/0.8194 | 26.96/0.8257 | 30.20/0.8605 | 24.17/0.7589 |
| SCGLANet[34] | ×4 | 25.29M | 27.03/0.8154 | 26.18/0.8052 | 30.23/0.8627 | 27.10/0.8209 | 26.87/0.8263 | 30.04/0.8568 | 24.30/0.7657 |
| Steformer[19] | ×4 | 1.34M | 26.61/0.8037 | 25.74/0.7906 | 29.29/0.8424 | 26.70/0.8098 | 26.45/0.8134 | 29.38/0.8425 | 23.58/0.7376 |
| SwinFIRSSR[4] | ×4 | 24.09M | 27.06/0.8175 | 26.15/0.8062 | 30.33/0.8676 | 27.16/0.8235 | 26.89/0.8283 | 30.44/0.8687 | 24.29/0.7681 |
| MSSFNet[32] | ×4 | 1.82M | 26.88/0.8098 | 26.07/0.7990 | 29.67/0.8498 | 26.97/0.8158 | 26.82/0.8219 | 29.77/0.8502 | 23.99/0.7508 |
| **DIFFSSR-T (Ours)** | ×4 | 1.69M | 26.90/0.8132 | 26.05/0.8021 | 29.88/0.8581 | 27.00/0.8191 | 26.78/0.8245 | 30.00/0.8589 | 24.04/0.7569 |
| **DIFFSSR (Ours)** | ×4 | 20.01M | **27.15/0.8201** | **26.23/0.8092** | **30.55/0.8723** | **27.25/0.8261** | **26.98/0.8312** | **30.64/0.8729** | **24.40/0.7722** |

Table 2: Comparison of perceptual quality on the real-world dataset StereoWeb20 [35].

| Dataset | Metric | NAFSSR [6] | SCGLANet-GAN [34] | DIFFSSR-GAN (Ours) |
|---|---|---|---|---|
| StereoWeb20 [35] | NIQE↓ | 5.7363 | 4.4174 | **4.1831** |
| | MANIQA↑ | 0.4648 | 0.5761 | **0.6184** |
| | MUSIQ↑ | 46.78 | 62.18 | **63.67** |
| | CLIPIQA↑ | 0.5144 | 0.6331 | **0.6857** |

we employ a Charbonnier L1 loss function [28] to measure the difference between the super-resolved and ground-truth stereo images. The model is trained on four NVIDIA GeForce RTX 3090 GPU, and the training process is carefully monitored to ensure convergence and to adjust hyperparameters as necessary for optimal performance.

## 4.2 Comparisons with State-of-the-art Methods

**Quantitative Evaluations.** Table 1 presents a comprehensive quantitative comparison of various methods for stereo image super-resolution (StereoSR) on the KITTI 2012, KITTI 2015, Middlebury, and Flickr1024 datasets. The metrics used for evaluation are the Peak Signal-to-Noise Ratio (PSNR) and Structural Similarity Index (SSIM), which are standard measures for assessing the quality of super-resolved images. The table reports these values for both individual left images and the average of left and right images in stereo pairs, providing a holistic view of performance across different scales. SwinFIRSSR, an extension of SwinFIR, incorporates Fast Fourier Convolution (FFC) to enhance the model's ability to capture global information, achieving state-of-the-art (SOTA) performance in StereoSR. However, as indicated in Table 1, our method, DIFFSSR, outperforms SwinFIRSSR across multiple benchmark datasets, including KITTI 2012, KITTI 2015, Middlebury, and Flickr1024, for both ×2 and ×4 scaling factors. Specifically, at a ×2 scale, DIFFSSR achieves a PSNR of 30.27dB and SSIM of 0.9301 on the Flickr1024 dataset, surpassing SwinFIRSSR's 30.14dB and 0.9286, respectively. This trend continues at a ×4 scale, where DIFFSSR garners a PSNR of 24.40dB and an SSIM of 0.7722, compared to SwinFIRSSR's 24.29dB and 0.7681. DIFFSSR demonstrates its superiority over NAFSSR-L as well. At a scale of ×2, NAFSSR-L achieves a PSNR of 31.45dB and SSIM of 0.9261 on the KITTI 2012 dataset, which is lower than DIFFSSR's scores. This gap widens at a scale of ×4, where DIFFSSR's PSNR and SSIM on the Middlebury dataset are

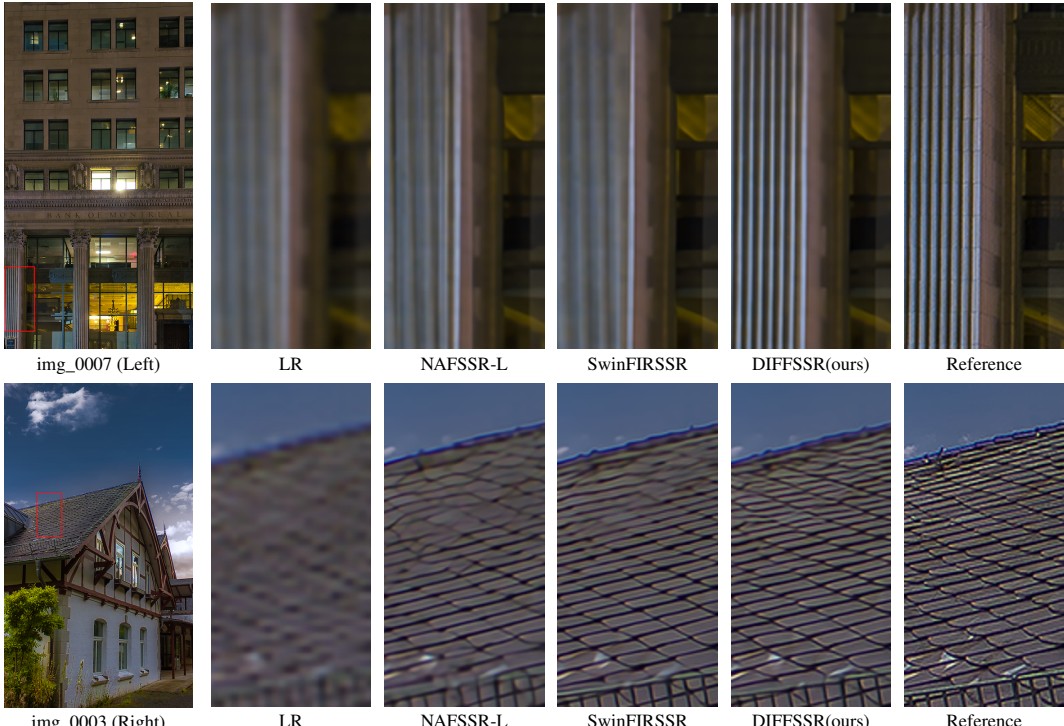

| img_0007 (Left) | LR | NAFSSR-L | SwinFIRSSR | DIFFSSR(ours) | Reference |

| img_0003 (Right) | LR | NAFSSR-L | SwinFIRSSR | DIFFSSR(ours) | Reference |

Figure 4: Visual results (×4) achieved by different methods on the Flickr1024 dataset.

significantly higher than NAFSSR-L's, 30.55dB vs. 30.11dB and 0.8723 vs. 0.8601, respectively. These results underscore the superior performance of DIFFSSR in reconstructing high-quality images from degraded stereo pairs. Another significant advantage of DIFFSSR is its parameter efficiency. As indicated in Table 1, DIFFSSR has 19.86M parameters at a ×2 scale and 20.01M parameters at a ×4 scale, which is considerably fewer than SwinFIRSSR's 23.94M and 24.09M parameters, respectively. This parameter reduction does not compromise performance; in fact, it enhances it.

**Visual Comparison.** Figure 4 presents a qualitative assessment of the stereo image super-resolution performance of three methods: NAFSSR, SwinFIRSSR, and our proposed DIFFSSR. The restoration of texture details is a critical aspect of super-resolution tasks, as fine details are often the first to be lost during the downscaling process. Figure 4 demonstrates that our DIFFSSR method outperforms both NAFSSR and SwinFIRSSR in this regard. The images produced by DIFFSSR exhibit sharper and more refined textures compared to the other two methods. In the visual results, it is evident that NAFSSR, while effective in enhancing image resolution, tends to blur finer details, particularly in areas with complex textures. SwinFIRSSR shows some improvement over NAFSSR in preserving details, but it still falls short when compared to DIFFSSR. The images generated by SwinFIRSSR occasionally exhibit a loss of fine structures, suggesting that traditional Transformers may have issues with attention noise, leading to the aggregation of a large amount of irrelevant information, which results in suboptimal reconstruction and over-smoothing problems. In contrast, DIFFSSR demonstrates a remarkable ability to recover even the most subtle texture details. This is attributed to the differential attention mechanism within the Diff Cross Attention Block (DCAB), which effectively differentiates between relevant and irrelevant context, thereby amplifying attention to important features and canceling out noise.

The superiority of our method can be attributed to the innovative design of the DCAB and SSCAM. The DCAB's differential attention mechanism is adept at distinguishing between relevant and irrelevant context, thus reducing attention noise. This, in turn, prevents over-smoothing by ensuring that the model focuses on significant features rather than dispersing attention across less relevant areas. The SSCAM, with its sliding window approach, enables more localized and adaptive feature integration, which is crucial for preserving texture details and preventing over-smoothing.

**Real-World Evaluation.** As shown in Table 2, our DIFFSSR-GAN consistently achieves the best perceptual quality across all metrics on the real-world dataset StereoWeb20. Specifically, it attains the lowest NIQE (4.1831) and the highest MANIQA (0.6184), MUSIQ (63.67), and CLIPIQA (0.6857),

outperforming both NAFSSR and SCGLANet-GAN by a notable margin. It is worth noting that SCGLANet-GAN was trained for 400,000 iterations with a batch size of 3 on 8 GPUs, whereas our DIFFSSR-GAN was trained for the same number of iterations but with a smaller batch size of 2 on only 4 GPUs, demonstrating better efficiency and robustness under more constrained computational resources. Moreover, our dataset simulation process follows the same degradation model as Real-ESRGAN, ensuring realistic noise and blur patterns that closely mimic real-world stereo imaging conditions. These results collectively validate that DIFFSSR-GAN effectively generalizes to real-world scenarios and achieves superior perceptual quality with less computational cost.

Table 3: Comparison results under different window size in Sliding Windows Stereo Cross-Attention Module.

| Window Size | Flickr1024 | KITTI 2012 | KITTI 2015 | Middlebury |
|---|---|---|---|---|
| - | 24.35 | 27.23 | 26.94 | 30.57 |
| 3 | 24.39 | 27.24 | 26.97 | 30.64 |
| 5 | 24.40 | 27.25 | 26.98 | 30.64 |
| 7 | 24.42 | 27.27 | 27.00 | 30.65 |

Table 4: The comparison results with Swin-FIRSSR. DIFF denotes replacing its Transformer with the DIFF Transformer.

| Method | Flickr1024 | KITTI 2012 | KITTI 2015 | Middlebury |
|---|---|---|---|---|
| SwinFIRSSR | 24.29 | 27.16 | 26.89 | 30.44 |
| DIFF | 24.11 | 26.99 | 26.71 | 30.14 |
| DIFFSSR | 24.40 | 27.25 | 26.98 | 30.64 |

Table 5: Comparison results under different Positional Encoding (PE) and Feed-Forward Neural Network (FFN). RoPE [21] and RPE [7] denote Rotary Position Embedding and Relative Positional Encoding, respectively. Depth denotes $0.8 - 0.6 * e^{-0.3*depth}$.

| Method | Factor | Flickr1024 | KITTI 2012 | KITTI 2015 | Middlebury |
|---|---|---|---|---|---|
| | - | 24.10 | 26.96 | 26.69 | 30.13 |
| PE | RoPE | 24.11 | 26.99 | 26.71 | 30.14 |
| | RPE | 24.13 | 26.99 | 26.72 | 30.17 |
| FFN | SwiGLU | 24.11 | 26.99 | 26.71 | 30.14 |
| | MLP | 24.18 | 27.06 | 26.85 | 30.31 |
| | 0.8 | 24.11 | 26.99 | 26.71 | 30.14 |
| $\lambda_{init}$ | Depth | 24.11 | 26.99 | 26.71 | 30.12 |
| | Learnable | 24.11 | 26.99 | 26.71 | 30.14 |

## 4.3 Ablation Study

### 4.3.1 Impact of the Window Sizes

The ablation study results presented in Table 3 offer valuable insights into the impact of varying window sizes within the Sliding Stereo Cross-Attention Module (SSCAM). It is observed that when the window size is increased from 1 to 3, there is a significant and consistent improvement in PSNR across test datasets. This suggests that a moderate window size of 3 already captures additional contextual information beneficial for super-resolution tasks. Further increasing the window size to 5 results in marginal improvements over the window size of 3, with PSNR values of 24.40dB, 27.25 dB, 26.98 dB, and 30.64 dB on Flickr1024, KITTI 2012, KITTI 2015, and Middlebury datasets, respectively. While the window size of 7 yields the highest PSNR values across datasets, its excessive computational requirements make it impractical. It is worth noting that odd-numbered window sizes (1, 3, 5, 7) are adopted due to their inherent symmetry and implementation simplicity, which are common and reasonable design choices in visual models. The window size of 3 provides a near-optimal balance, offering significant performance gains with reasonable computational costs, making it the chosen configuration for our model.

### 4.3.2 Impact of the DIFF Transformer

The ablation study in Table 4 reveals that directly replacing the Transformer in SwinFIRSSR with the DIFF Transformer (referred to as the DIFF) leads to a decrease in PSNR across all datasets, indicating that the DIFF Transformer does not translate its benefits from language tasks to visual tasks effectively. In contrast, our proposed DIFFSSR, which incorporates a redesigned DIFF Transformer architecture tailored for stereo super-resolution, outperforms both SwinFIRSSR and the DIFF methods. This enhancement underscores the importance of architecture design in leveraging the complementary information present in stereo image pairs and highlights the potential of the DIFF Transformer when adapted appropriately for visual tasks.

### 4.3.3 Impact of Some Factors in Transformer

The ablation study presented in Table 5 provides a detailed analysis of the impact of different Positional Encodings (PE) and Feed-Forward Neural Network (FFN) architectures on the performance of the DIFF Transformer model. The first group of rows in the table compares the performance of three different positional encoding methods. The results indicate that positional encoding consistently improves performance across all datasets. However, the analysis also suggests that while RoPE has shown significant benefits in LLMs, it may not effectively capture the spatial relationships in visual data, as indicated by the relatively small improvement in performance with RoPE compared to RPE. This aligns with prior findings suggesting that RoPE may not be the most suitable for visual tasks that rely heavily on relative spatial relationships, where RPE offers a more fitting approach. The second group of rows compares the performance of two different FFN architectures. MLP outperforms SwiGLU across all datasets, yielding improvements of 0.07, 0.07, 0.14, and 0.17 for the respective datasets. This suggests that MLP is more suited for this task, potentially due to its ability to model more complex non-linearities and interactions between features, which may be particularly beneficial in tasks that require higher levels of abstraction or richer feature representations.

## 5 Conclusion

In this paper, we introduced DIFFSSR, a novel approach to stereo image super-resolution that leverages the strengths of the Differential Transformer to address the limitations of traditional Transformer models. Through the innovative design of the Diff Cross Attention Block (DCAB) and the Sliding Stereo Cross-Attention Module (SSCAM), DIFFSSR effectively reduces attention noise and improves the integration of cross-view information, leading to superior performance in recovering fine texture details and preventing over-reconstruction. Our method achieved state-of-the-art results across multiple benchmark datasets, outperforming existing methods such as NAFSSR and SwinFIRSSR. The visual comparison clearly demonstrates DIFFSSR's ability to restore low-resolution images with greater accuracy and detail, while also avoiding the over-smoothing artifact that plague traditional Transformer-based models. The success of DIFFSSR highlights the importance of tailored neural network architectures for specific visual tasks and the potential of Differential Transformer in improving the performance of super-resolution models.

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

# A  Additional Ablation Study

### A.0.1  The Effectiveness of SSCAM

As shown in Table 6, for the ablation study of SSCAM, we replaced the SSCAM with SCAM in the Diff Cross Attention Layer. The experimental results demonstrate that our proposed SSCAM achieves superior performance compared to SCAM. To further validate the effectiveness of SSCAM, we simulated misalignment between left and right views by randomly applying vertical shifts of 0-3 pixels to the right view in the Flickr1024 validation dataset. We evaluated only the PSNR of the left view in this setup. The results indicate that our SSCAM incurs negligible performance degradation, while the methods based on SCAM (SwinFIRSSR and NAFSSR) exhibit a significant drop in performance.

Table 6: The effectiveness of our SSCAM.

| Method | DIFFSSR | DIFFSSR(SCAM ) | SwinFIRSSR | NAFSSR |
|---|---|---|---|---|
| Flickr1024 | 24.34 | 24.27 | 24.22 | 24.11 |
| Flickr1024 + vertical shift | 24.31(-0.03) | 24.05(-0.22) | 23.98(-0.24) | 23.74(-0.37) |

### A.0.2  The Visualization of the Attention Map

As shown in Figure 5, through the visualization of attention maps, it can be observed that self-attention mechanisms tend to assign higher weights to irrelevant pixels, while differential attention (our) demonstrates the capability to effectively mitigate noise interference. Furthermore, comprehensive theoretical analyses and experimental validations of the noise suppression capacity inherent in differential attention have been rigorously demonstrated by the original authors.

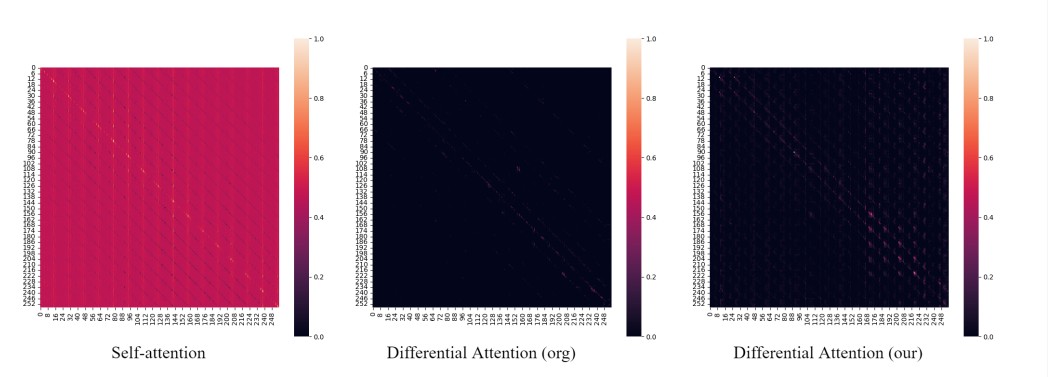

Figure 5: The visualization of the attention map.

### A.0.3  Impact of the Window Sizes

The Table 7 presents a comprehensive analysis of the impact of varying window sizes on the performance of the Sliding Stereo Cross-Attention Module in stereo image super-resolution. The results indicate that as the window size increases from 1 to 7, there is a corresponding improvement in performance such as PSNR and SSIM across different datasets. Specifically, the largest window size of 7 yields the best performance, with PSNR values of 24.42dB, 27.27dB, 27.00dB, and 30.65dB respectively. However, this enhancement in performance comes at a cost. The computational complexity, measured in GFlops, and the inference time both increase significantly with larger window sizes. For example, while a window size of 1 has a computational cost of 137.42 GFlops and an inference time of 0.1003 seconds, these values rise to 435.54 GFlops and 0.1683 seconds for a window size of 7. Despite the superior performance of the largest window size, the study concludes that a window size of 3 strikes an optimal balance between performance and computational efficiency. This size offers a reasonable trade-off, with a moderate increase in computational cost (187.11 GFlops) and inference time (0.1274 seconds) compared to the baseline, while still achieving notable performance gains. This makes it a practical choice for real-world applications where both performance and efficiency are critical considerations.

Table 7: Comparison results under different window size in Sliding Windows Stereo Cross-Attention Module. We test our method on NVIDIA GeForce RTX 3090 GPU with the resolution $32 \times 96$.

| Window Size | $\#P$ | $GFlops$ | $Time$ | Flickr1024 | KITTI 2012 | KITTI 2015 | Middlebury |
|---|---|---|---|---|---|---|---|
| - | 20.01 | 137.42 | 0.1003s | 24.35 | 27.23 | 26.94 | 30.57 |
| 3 | 20.01 | 187.11 | 0.1274s | 24.39 | 27.24 | 26.97 | 30.64 |
| 5 | 20.01 | 286.48 | 0.1442s | 24.40 | 27.25 | 26.98 | 30.64 |
| 7 | 20.01 | 435.54 | 0.1683s | 24.42 | 27.27 | 27.00 | 30.65 |

## A.0.4  Training Free for the Sliding Stereo Cross-Attention Module (SSCAM)

The experimental results presented in Table 8 demonstrate the critical advantage of the proposed SSCAM in achieving training-free adaptability to varying window sizes during inference, a capability lacking in the NAFSSR's SCAM. Specifically, when trained with a window size of 1 and tested with larger window sizes (3 or 5), SSCAM-based DIFFSSR consistently improves performance across all datasets, whereas NAFSSR suffers significant degradation. For instance, on the Flickr1024 dataset, NAFSSR's PSNR drops from 24.16 dB (test window=1) to 24.13 dB (test windows=3), indicating its sensitivity to window size mismatches. In contrast, DIFFSSR trained with window=1 exhibits a performance gain from 24.35 dB to 24.37 dB when tested with larger windows, highlighting SSCAM's intrinsic ability to exploit expanded spatial contexts without retraining. Similar trends are observed on KITTI and Middlebury datasets, where DIFFSSR maintains or slightly enhances accuracy regardless of test window sizes, while NAFSSR's scores consistently decline. Furthermore, DIFFSSR exhibits remarkable robustness even when trained with larger windows (3 or 5). For example, DIFFSSR trained with window=5 achieves 24.40 dB on Flickr1024 when tested with window=5, marginally outperforming its window=3 training counterpart (24.39 dB). This suggests SSCAM's bidirectional compatibility—it not only enables upward window size generalization from smaller training windows but also maintains stability when trained with larger windows, such versatility is absent in NAFSSR.

Table 8: Comparison results under different window size in Sliding Windows Stereo Cross-Attention Module.

| Method | Train Window Size | Test Window Size | Flickr1024 | KITTI 2012 | KITTI 2015 | Middlebury |
|---|---|---|---|---|---|---|
| NAFSSR | 1 | 1 | 24.16 | 27.11 | 26.94 | 30.19 |
|  |  | 3 | 24.14 | 27.09 | 26.92 | 30.18 |
|  |  | 5 | 24.13 | 27.07 | 26.91 | 30.16 |
| DIFFSSR | 1 | 1 | 24.35 | 27.23 | 26.94 | 30.59 |
|  |  | 3 | 24.37 | 27.23 | 26.97 | 30.59 |
|  |  | 5 | 24.37 | 27.23 | 26.97 | 30.59 |
| DIFFSSR | 3 | 1 | 24.37 | 27.24 | 26.96 | 30.61 |
|  |  | 3 | 24.39 | 27.24 | 26.97 | 30.64 |
|  |  | 5 | 24.39 | 27.24 | 26.97 | 30.64 |
| DIFFSSR | 5 | 1 | 24.36 | 27.24 | 26.96 | 30.60 |
|  |  | 3 | 24.39 | 27.25 | 26.97 | 30.64 |
|  |  | 5 | 24.40 | 27.25 | 26.98 | 30.64 |

## A.0.5  Parameters vs. PSNR

The Figure 6 provides a comparison of various stereo image super-resolution methods, highlighting the trade-off between the number of parameters and PSNR performance on the Flickr1024 dataset. Our methods, DIFFSSR and DIFFSSR-T, demonstrate significant improvements in performance and parameter efficiency. Specifically, DIFFSSR achieves SOTA performance with a PSNR of 24.40dB ($\times 4$), surpassing other methods including NAFSSR-L (24.17 dB) and SwinFIRSSR (24.29 dB). Notably, DIFFSSR requires fewer parameters than both NAFSSR-L [6] and SwinFIRSSR [4]. At a $\times 4$ scale, DIFFSSR has 20.01M parameters versus 23.83M for NAFSSR-L and 24.09M for Swin-FIRSSR, representing reductions of 16.0% and 16.9%, respectively. DIFFSSR-T achieves comparable performance to NAFSSR-B but requiring fewer parameters (1.69M), 75.1% reduction compared to NAFSSR-B's 6.80M. Additionally, DIFFSSR-T surpasses NAFSSR-S (1.56M parameters) in PSNR (24.04 dB vs. 23.88 dB), demonstrating a 0.16 dB improvement despite similar parameter counts.

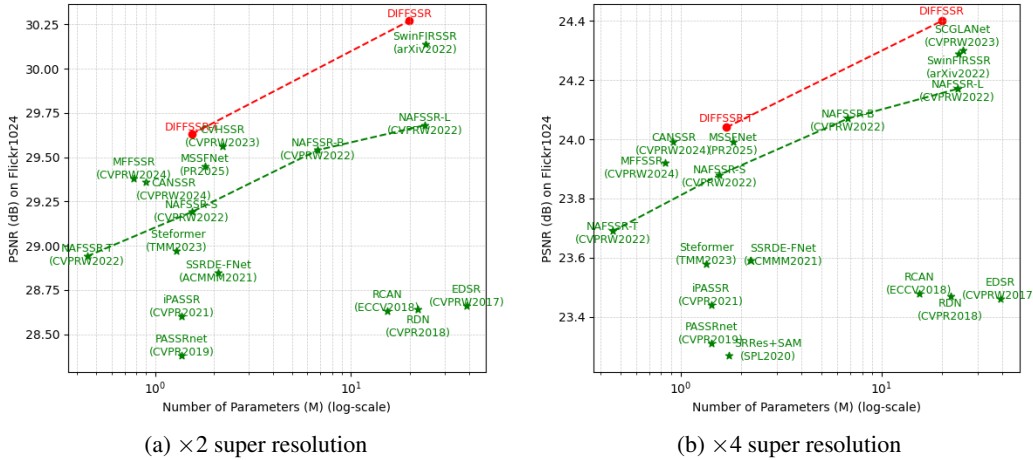

(a) ×2 super resolution          (b) ×4 super resolution

Figure 6: The trade-off between the number of parameters and PSNR on the Flickr1024 dataset.

### A.0.6 The Comparison with Some More Recent SOTA Methods.

The Figure 6 also provides a comparison of various recent stereo image super-resolution methods. Although SwinFIRSSR [4] was initially proposed in 2022, it has consistently maintained SOTA performance. Therefore, we select SwinFIRSSR as the primary baseline for comparison. According to the NTIRE 2023 Challenge on Stereo Image Super-Resolution [36] results, SwinFIRSSR won the championship in Track 2. Furthermore, the NTIRE 2024 Challenge on Stereo Image Super-Resolution [37] results demonstrate that the tiny version of SwinFIRSSR achieved dual championships in both Track 1 and Track 2. In contrast, other methods exhibited inferior performance: CANSSR [38] ranked 6th in Track 1, MFFSSR [39] ranked 7th in Track 1 and 9th in Track 2. Notably, our proposed DIFFSSR surpasses SwinFIRSSR across all four benchmark datasets, which provides compelling evidence for the effectiveness of our methodology. Our DIFFSSR also achieves higher performance than them.

## B  Limitations and Future Work

This work represents the first exploration of the Differential Transformer in visual tasks, specifically focusing on stereo image super-resolution (StereoSR). While the proposed DIFFSSR achieves state-of-the-art performance in StereoSR, its current design is tailored exclusively for this specific task. The adaptability and generalizability of the Differential Transformer to other visual domains (e.g., image classification, object detection, semantic segmentation) remain unexplored. Additionally, the computational complexity of the Sliding Stereo Cross-Attention Module (SSCAM) increases with larger window sizes, which may affect real-time applications despite the optimal balance achieved with a window size of 3.

Future work will focus on generalizing the DIFF Transformer framework to diverse vision tasks. For instance, investigating its potential in image classification by adapting the differential attention mechanism to hierarchical feature learning, or exploring its use in dense prediction tasks like detection and segmentation, could validate its broader utility.

## C  Additional Visual Results

In this part, we provide additional visual results compared to the SOTA method.

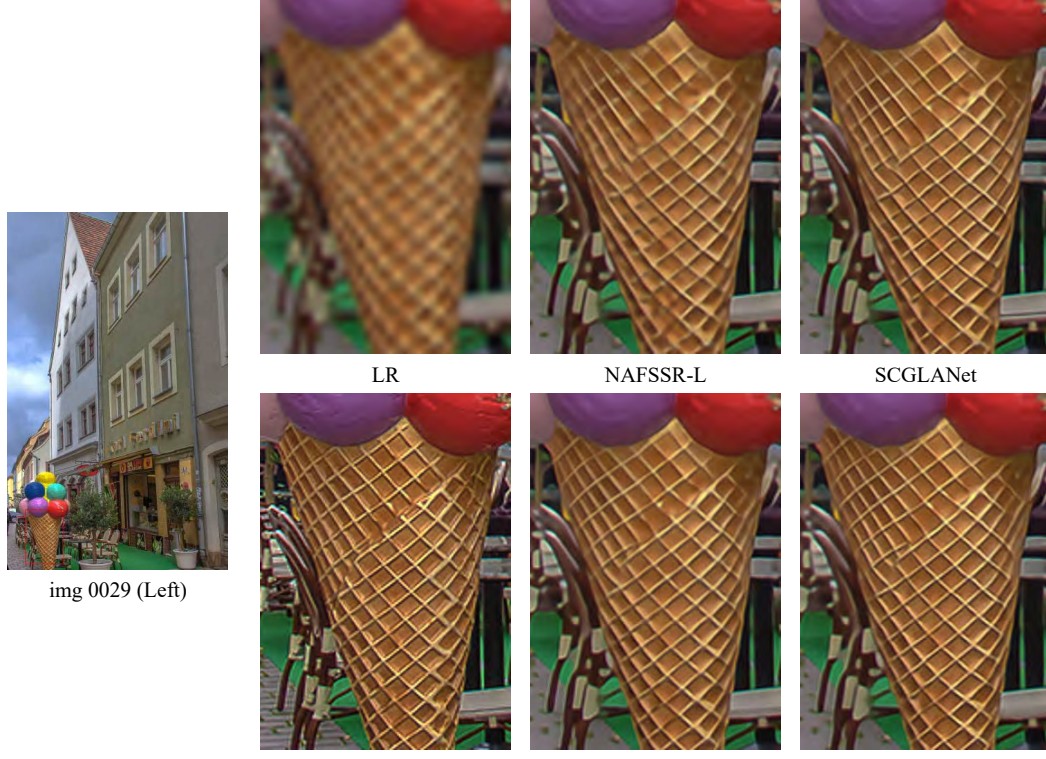

img 0029 (Left)

| | | |
|---|---|---|
| LR | NAFSSR-L | SCGLANet |
| Reference | SwinFIRSSR | DIFFSSR(ours) |

Figure 7: Visual results (×4) achieved by different methods on the Flickr1024 dataset.

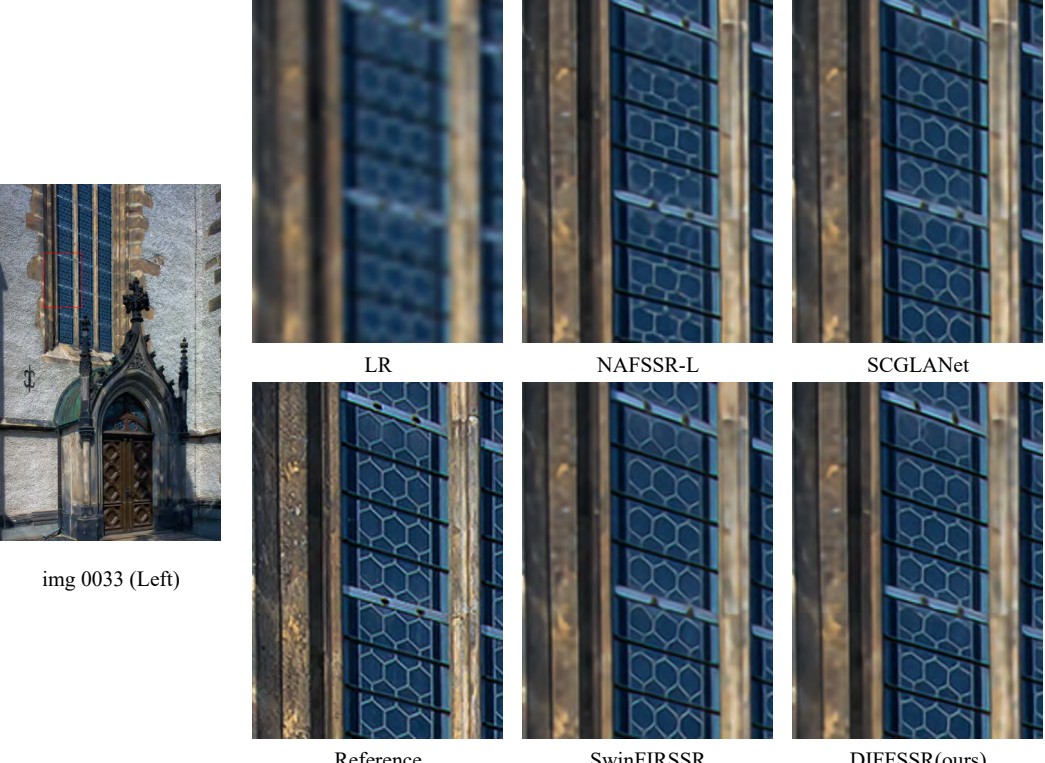

img 0033 (Left)

| | | |
|---|---|---|
| LR | NAFSSR-L | SCGLANet |
| Reference | SwinFIRSSR | DIFFSSR(ours) |

Figure 8: Visual results (×4) achieved by different methods on the Flickr1024 dataset.

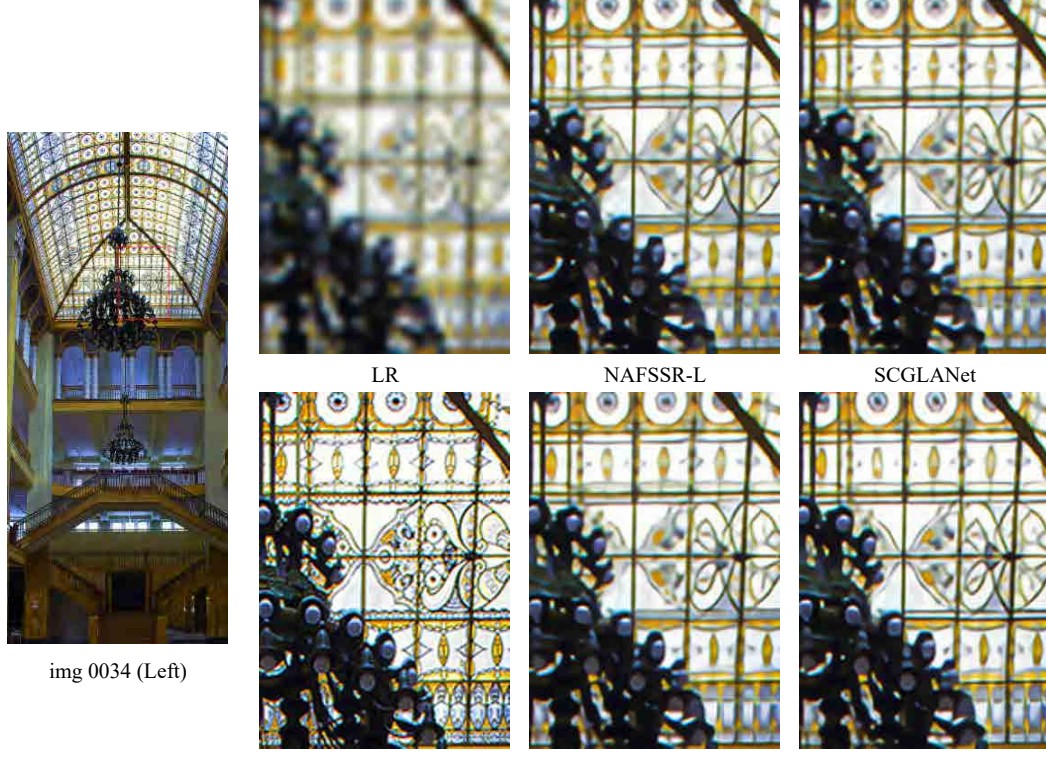

img 0034 (Left)

Figure 9: Visual results (×4) achieved by different methods on the Flickr1024 dataset.

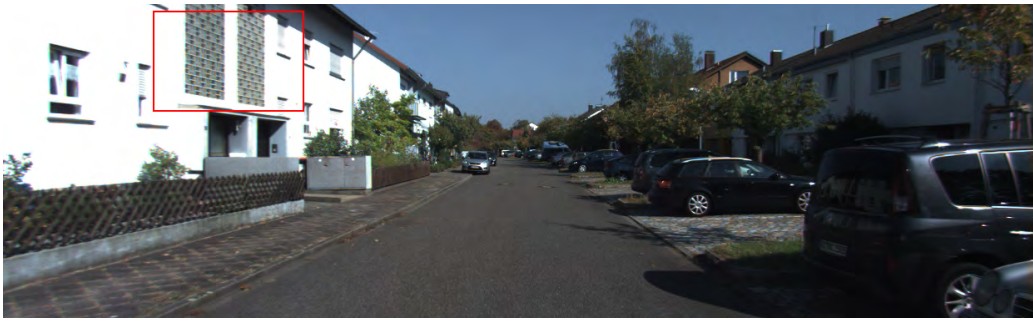

img 000000 (Left)

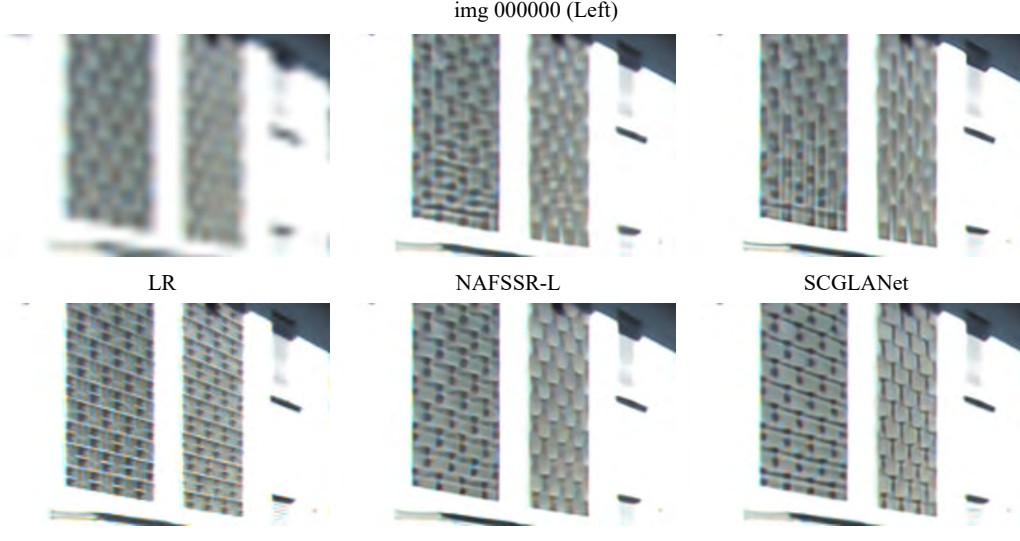

Figure 10: Visual results (×4) achieved by different methods on the KITTI2012 dataset.

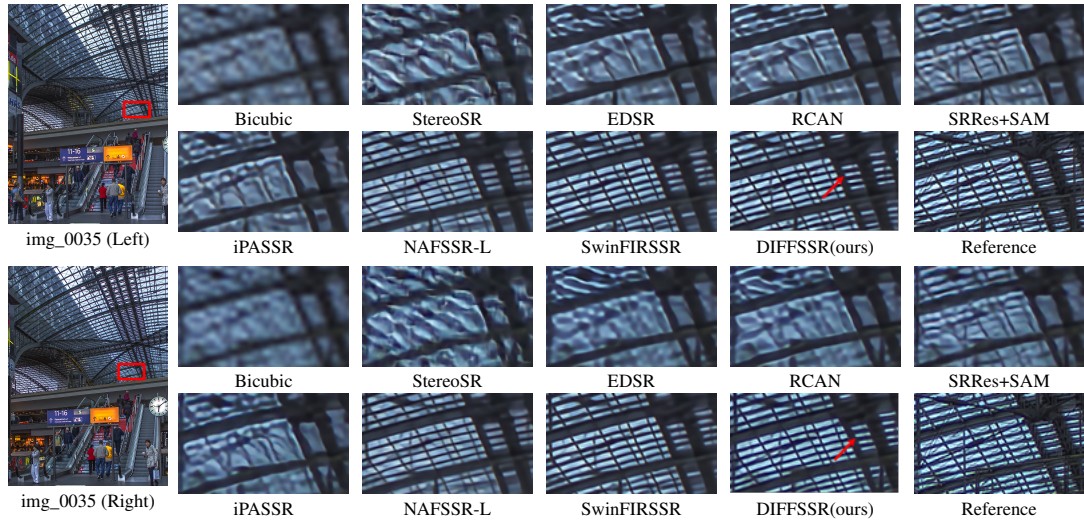

Figure 11: Visual results (×4) achieved by different methods on the Flickr1024 dataset.

