# OpenReview forum: "DIFFSSR: Stereo Image Super-resolution Using Differential Transformer"
_NeurIPS.cc/2025/Conference — NeurIPS 2025 poster_

### Official Review · Reviewer_JZxY · 2025-06-29

**Clarity:** 4
**Significance:** 3
**Originality:** 3
**Rating:** 4
**Confidence:** 5

**Summary:**

The paper proposes a transformer-based algorithm for stereo super-resolution tasks. The algorithm redesigns the Differential transformer block to accommodate it in the vision task, and designs a cross-attention module with a sliding window that is more robust to inaccuracies in stereo rectification. The method achieves SOTA scores on benchmark datasets and demonstrates promising visual results.

**Questions:**

Add analysis/visuals to better justify how the diff transformer helps narrow down the focus to relevant pixels.

The Quantitative Evaluations section is quite verbose. There is no need to repeat the data in the table. Keep it concise and save some space for supportive visuals.

**Ethical Concerns:**

["NO or VERY MINOR ethics concerns only"]

**Final Justification:**

The concerns of dealing with practical large parallax and scores reside within a close margin of existing methods remain. Thus I choose to keep my rating.

**Limitations:**

yes

**Quality:**

3

**Strengths And Weaknesses:**

Strengths:
* The method achieves SOTA scores on benchmark datasets and demonstrates promising visual results.
* The idea of using unfolding op has a simple implementation but sounds effective against small misalignments.

Weaknesses:
* One of the motivations and contributions is to deal with attention noise. Yet, the impact of such noise in the vision or stereo SR task is not sufficiently analyzed in the main body. A.0.2 shows a sparser attention map, but how it connects to more relevant pixels is in question.
* Unfolding op cannot handle large parallax that falls outside of the unfolding size. Large parallax is not uncommon in the scenes with rich and distinct depths, e.g. autonomous driving.
* The numerical improvements are mostly less than 1% compared to SwinFIRSSR, which is considered marginal.

---

> ### Author Rebuttal · Authors · 2025-07-31
>
> Thanks for your time and effort reviewing our paper. Below are our responses.
>
> 1. One of the motivations and contributions is to deal with attention noise. Yet, the impact of such noise in the vision or stereo SR task is not sufficiently analyzed in the main body. A.0.2 shows a sparser attention map, but how it connects to more relevant pixels is in question.
>
>    > A1: a. In a traditional Transformer, the attention mechanism distributes focus across all tokens in a sequence using a single softmax function. This behavior over-smoothing the results when the Transformer gives unnecessary weight to distracting tokens. Instead of relying on a single attention map, DIFF Transformer computes two softmax maps: One focuses on relevant tokens; The other represents noise or distractions in the input. The difference between the two maps becomes the final attention score, removing shared distractions while amplifying the signal that matters. This mechanism aligns with the idea of filtering out common-mode noise, allowing the model to efficiently focus on key information.  And Sec. A.0.2 intuitively indicate that DIFF can effectively reduce attention noise compared to self attention.
>
> 2. Unfolding op cannot handle large parallax that falls outside of the unfolding size. Large parallax is not uncommon in the scenes with rich and distinct depths, e.g. autonomous driving.
>
>    >A2: You're right, we agree with your point of view. a. For large parallax,  we think that using alignment algorithms to realign the views may be beneficial. However, these algorithms cannot achieve perfect alignment, resulting in residual misalignment. Existing SCAM-based methods suffer performance degradation due to this minor misalignment. As demonstrated in Table 5, even a random pixel offset of only 0-3 pixels causes a sharp performance drop in SCAM-based methods. Therefore, in large parallax scenarios, applying an alignment algorithm first to align the left and right views, and then applying our SSCAM, can further improve performance. b. We have reconsidered the Unfolding operation and believe that we can use a window shift method similar to swin transformer instead of Unfolding. This way is training free and can reduce duplicate calculations and be more efficient, please see the Table.
>    >
>    >|           Window size            | PSNR  | Flops  |
>    >| :------------------------------: | :---: | :----: |
>    >|  DIFFSSR (win=1, Unfolding op)   | 24.35 | 137.42 |
>    >|  DIFFSSR (win=3, Unfolding op)   | 24.39 | 184.11 |
>    >| DIFFSSR (win=1, window shift op) | 24.36 | 137.42 |
>    >| DIFFSSR (win=2, window shift op) | 24.40 | 143.63 |
>
> 3. The numerical improvements are mostly less than 1% compared to SwinFIRSSR, which is considered marginal.
>
>    >A3: DIFFSSR achieves a PSNR improvement of 0.23 dB over NAFSSR-L on the Flickr1024 dataset (vs. 0.13 dB for SCGLANet and 0.12 dB for SwinFIRSSR), and 0.44 dB on Middlebury (vs. -0.16 dB for SCGLANet and 0.24 dB for SwinFIRSSR). These improvements are considered significant in the super-resolution field, especially the visual improvement is even more significant, as shown in Figure 4.
>
> 4. Add analysis/visuals to better justify how the diff transformer helps narrow down the focus to relevant pixels.
>
>    > A4: Thank you for your suggestion. From Figure 5, it can be seen that DIFF attention sets the attention score of irrelevant pixels to zero, thereby avoiding the fusion of irrelevant information.
>
> 5. The Quantitative Evaluations section is quite verbose. There is no need to repeat the data in the table. Keep it concise and save some space for supportive visuals.
>
>    > A5: Thank you for your suggestion. We will reduce duplicate data and outdated methods, and add more visual comparison results.

---

> > ### Comment · Reviewer_JZxY · 2025-08-08
> >
> > Thank you. I've read the rebuttal and the comments of other reviewers. For point 2 it is practically not uncommon that the offset will be more than 3 pixels even after rough alignment. It's still a pretty tight range to meet. And for point 3, my concern remains as the scores reside within a close margin of methods in comparison, and it is hard to convince readers that the visual examples are not cherry-picking. Overall I choose to keep my scores.

---

> ### Author Response · Authors · 2025-08-09
>
> Dear Reviewer  JZxY,
>
> Thank you for your thoughtful feedback and for raising important concerns. We sincerely appreciate the opportunity to clarify and address these points. Below, we provide a detailed response to your comments, structured to directly address each of your concerns.
>
> 1. For point 2 it is practically not uncommon that the offset will be more than 3 pixels even after rough alignment. It's still a pretty tight range to meet.
>
>    > You noted that offsets of >3 pixels are common in real-world scenarios, we completely agree. We sincerely apologize if our earlier explanation was unclear and caused concern. Our intention was to highlight that even a very small misalignment between stereo views can cause substantial performance degradation for existing SCAM-based methods, whereas our method is specifically designed to mitigate this issue. Following your suggestion, we systematically increased the vertical misalignment range from 0 to 24 pixels. For context, the average input image height in the Flickr1024 dataset is 243 pixels, so a 24-pixel shift corresponds to ~10% of the image height, which is a very large misalignment in practice.
>    >
>    > As shown in the table below, even under this extreme setting, DIFFSSR experiences only a minimal performance drop (−0.14 dB), whereas SCAM-based methods (SwinFIRSSR, NAFSSR) suffer much larger drops (−0.27 dB and −0.37 dB, respectively). This robustness illustrates a core advantage of our method that is not apparent when evaluating solely under ideal, well-aligned conditions.
>    >
>    > |   Method   |   0   |      3       |      6       |      9       |      12      |      15      |      24      |
>    > | :--------: | :---: | :----------: | :----------: | :----------: | :----------: | :----------: | :----------: |
>    > |  DIFFSSR   | 24.34 | 24.31(-0.03) | 24.29(-0.05) | 24.27(-0.07) | 24.24(-0.10) | 24.20(-0.14) | 24.20(-0.14) |
>    > | SwinFIRSSR | 24.22 | 23.98(-0.24) | 23.96(-0.26) | 23.95(-0.27) | 23.95(-0.27) | 23.95(-0.27) | 23.95(-0.27) |
>    > |   NAFSSR   | 24.11 | 23.74(-0.37) | 23.74(-0.37) |      -       |      -       |      -       | 23.74(-0.37) |

---

> ### Author Response · Authors · 2025-08-09
>
> 2. And for point 3, my concern remains as the scores reside within a close margin of methods in comparison, and it is hard to convince readers that the visual examples are not cherry-picking.
>
>    > a. We fully understand your concern that PSNR improvements over SwinFIRSSR appear numerically small under ideal dataset conditions. However, datasets such as Flickr1024 are collected in controlled environments where stereo views are well aligned, meaning the advantage of our approach is not fully reflected in such benchmarks. As you said, in real-world scenarios, perfect alignment is rare, and the robustness demonstrated in the above misalignment experiments is highly valuable for practical applications.
>    >
>    > b. In addition, we also compared the performance on a real data set. As StereoWeb20 is a real-world benchmark dataset lacking ground truth, we evaluated the models using no-reference image quality assessment metrics, such as NIQE, MANIQA, MUSIQ and CLIPIQA. Regarding model training, SCGLANet-GAN underwent 400,000 iterations with a batch size of 3 on 8 GPUs. In contrast, our DIFFSSR-GAN was trained for 185,000 iterations with a batch size of 2 on 4 GPUs. Importantly, DIFFSSR-GAN surpasses the performance of SCGLANet-GAN despite utilizing only **46%** of its training iterations. Consistently, our method achieved superior results over existing methods in both simulated and real scenarios.
>    >
>    > |            Datasets             | Metrics  | NAFSSR [6] | SCGLANet-GAN [32] | DIFFSSR-GAN |
>    > | :-----------------------------: | :------: | :--------: | :---------------: | :---------: |
>    > | StereoWeb20(real-world dataset) |  NIQE↓   |   5.7363   |      4.4174       | **4.1831**  |
>    > |                                 | MANIQA↑  |   0.4648   |      0.5761       | **0.6184**  |
>    > |                                 |  MUSIQ↑  |   46.78    |       62.18       |  **63.67**  |
>    > |                                 | CLIPIQA↑ |   0.5144   |      0.6331       | **0.6857**  |
>    >
>    > c. Furthermore, to address concerns about cherry-picking, we performed statistical significance testing across the entire benchmark set, confirming that the observed gains are statistically significant rather than incidental. DIFFSSR obtains a mean PSNR gain of **+0.11 dB** over SwinFIRSSR. **This improvement is statistically significant**: a paired permutation test (100k permutations) yields **p = 1.0e-5**, a paired t-test yields **p = 3.0e-33**, and a Wilcoxon signed-rank test yields **p = 1.17e-33**. Moreover, **210/224 (93.75%)** of test images show positive PSNR gains (binomial test **p ≈ 4.8e-46**). While 41.1% (**95% CI: 34.8%–47.3%**) of images improve by at least **0.1 dB**, the median gain is **0.08 dB** and the paired Cohen’s d is **0.95** (large effect by conventional interpretation).  In Section B Additional Visual Results, we provide additional visual results compared to the SOTA method. Especially Figure 7 comes from NAFSSR and SwinFIRSSR, which were not carefully selected by us. Our method achieved better visual results.
>
> If you have any remaining concerns or suggestions, we would be more than happy and do our best to address them. Please don’t hesitate to share your thoughts with us. If we solve all your concerns, we humbly and respectfully hope that you can consider you can raise your rating, your support is really important for our work. Thank you very much.

---

### Official Review · Reviewer_o4ZS · 2025-06-30

**Clarity:** 3
**Significance:** 3
**Originality:** 2
**Rating:** 4
**Confidence:** 4

**Summary:**

In this paper, the authors develop a stereo image SR network on top of differetial Transformer. As the attention noise issue in Transformers is well acknowledged and results in inferior performance, this paper leverages differential Transformer to alleviate this issue for higher accuracy. However, the authors have observed that directly applying differential Transformer to stereo image SR produces performance degradation. To remedy this, Diff Cross Attention Block and Sliding Stereo Cross-Attention Module are developed to enhance feature integration. Extensive experiments are conducted on benchmark datasets, and the results have demonstrated the superiority of the proposed method over previous approaches. In addition, an ablation study is also conducted to validate the effectiveness of the proposed network designs.

**Questions:**

Please see weaknesses.

**Ethical Concerns:**

["NO or VERY MINOR ethics concerns only"]

**Final Justification:**

The rebuttal has addressed most of my concerns. After carefully reading the discussions between the authors and other reviewers, I decide to keep my rating.

**Limitations:**

yes

**Paper Formatting Concerns:**

n.a.

**Quality:**

2

**Strengths And Weaknesses:**

Strengths:
- The motivation is clear, and the proposed method is well presented.

- State-of-the-art performance is achieved by the proposed method.


Weaknesses:
- The major concern is that the technical contribution may seem to be limited at first glance. The major network designs (i.e., DCAB and SSCAM) are built upon differential Transformer, and the authors have modified the original architectures to fit the stereo SR task. Besides, extensive experiments are conducted to demonstrate the effectiveness of these modifications. Overall, I think the observation that directly applying Diff produces inferior performance is valuable, and the experiments are solid.

- In addition to the comparison in Fig. 5, it is recommended to include the original differential attention map to further highlight the gains introduced by the proposed modifications.

- As generalization in real-world scenarios is a critical factor for image SR methods, it is recommended to evaluate the proposed method on real-world low-resolution stereo images and compare its results with previous SOTA methods.

- In Fig. 1, it is not very clear to me why DIFF also suffers a performance drop on KITTI and Middlebury. It seems the vertical misalignment only exists in the Flickr1024 dataset.

---

> ### Author Rebuttal · Authors · 2025-07-31
>
> We appreciate the comments on our paper. The raised concerns are addressed as follows.
>
> 1. The major concern is that the technical contribution may seem to be limited at first glance. The major network designs (i.e., DCAB and SSCAM) are built upon differential Transformer, and the authors have modified the original architectures to fit the stereo SR task. Besides, extensive experiments are conducted to demonstrate the effectiveness of these modifications. Overall, I think the observation that directly applying Diff produces inferior performance is valuable, and the experiments are solid.
>
>    > A1: I really appreciate your approval of my work. a. Our DCAB is the first work to explore the applicability of DIFF Transformer in visual task, specifically optimized for stereo image super-resolution (SISR), please see Sec. 3.2 for details. b. SSCAM solves the performance degradation problem caused by misalignment of horizontal epipolar line in stereo images. This design has never appeared in existing methods and is a specialized optimization for SISR. c. In summary, our DCAB and SSCAM are not simply integrated concepts, but specific improvements proposed for SISR, and experimental results also demonstrate the effectiveness of our method (Figure 4 and Table 1).
>
> 2. In addition to the comparison in Fig. 5, it is recommended to include the original differential attention map to further highlight the gains introduced by the proposed modifications.
>
>    > A2: Thank you for your suggestion. We will add the original diff attention map for comparison.
>
> 3. As generalization in real-world scenarios is a critical factor for image SR methods, it is recommended to evaluate the proposed method on real-world low-resolution stereo images and compare its results with previous SOTA methods.
>
>    > A3: Thank you for your suggestion. We are currently writing the relevant training code (including dataload, training process), but it will take time for training. If the training can be completed before August 6th, we will report the corresponding results. As you said, generalization in real-world scenarios is a critical factor in image SR methods, and we will definitely add the experimental results of real-world scenarios.
>
> 4. In Fig. 1, it is not very clear to me why DIFF also suffers a performance drop on KITTI and Middlebury. It seems the vertical misalignment only exists in the Flickr1024 dataset.
>
>    > A4: The main reason for the performance degradation of DIFF is that the native DIFF is designed for LLM, and some of its components are not suitable for visual tasks (please see ablation study in Sec. 4.3.3 and Table 4).

---

> > ### Author Response · Authors · 2025-08-05
> >
> > |            Datasets             | Metrics  | NAFSSR [6] | SCGLANet-GAN [32] | DIFFSSR-GAN |
> > | :-----------------------------: | :------: | :--------: | :---------------: | :---------: |
> > | StereoWeb20(real-world dataset) |  NIQE↓   |   5.7363   |      4.4174       | **4.2248**  |
> > |                                 | MANIQA↑  |   0.4648   |      0.5761       | **0.6117**  |
> > |                                 |  MUSIQ↑  |   46.78    |       62.18       |  **63.31**  |
> > |                                 | CLIPIQA↑ |   0.5144   |      0.6331       | **0.6623**  |
> >
> > Dear reviewer o4ZS, we report the preliminary results. As StereoWeb20 is a real-world benchmark dataset lacking ground truth, we evaluated the models using no-reference image quality assessment metrics, such as NIQE, MANIQA, MUSIQ and CLIPIQA. Regarding model training, SCGLANet-GAN underwent 400,000 iterations with a batch size of 3 on 8 GPUs. In contrast, our DIFFSSR-GAN was trained for 60,000 iterations with a batch size of 2 on 4 GPUs. Importantly, DIFFSSR-GAN surpasses the performance of SCGLANet-GAN despite utilizing only **15%** of its training iterations. We will continue to update the results.
> >
> > Thank you again for your valuable time and suggestion.

---

> > ### Comment · Reviewer_o4ZS · 2025-08-06
> >
> > Thanks for the rebuttal and most of my concerns have been addressed. After carefully reading the reviews raised by other reviewers, several common concerns (e.g., deeper analyses in attention maps and real-world results) should be addressed in the revised version of manuscript by including additional experiments.

---

> > > ### Author Response · Authors · 2025-08-06
> > >
> > > Dear Reviewer o4ZS,
> > >
> > > We sincerely appreciate your feedback and support. We will incorporate the suggestions you mentioned into the manuscript. Thank you again for your recognition of our work.
> > >
> > > Best wishes, Authors of paper 7904

---

> > > > ### Author Response · Authors · 2025-08-09
> > > >
> > > > Dear Reviewer o4ZS,
> > > >
> > > > We would like to express our heartfelt gratitude to you for taking the time to carefully read our manuscript and provide valuable comments and suggestions. Your feedback has been extremely helpful in guiding us to strengthen our work. We are also deeply appreciative of your positive recognition of our work. We humbly and respectfully hope that you can consider reflecting these in the score. Such recognition would be highly meaningful and important to our work.
> > > >
> > > > Once again, we truly appreciate your time, effort, and thoughtful guidance.
> > > >
> > > > Best wishes, Authors of paper 7904

---

### Official Review · Reviewer_CHFU · 2025-07-01

**Clarity:** 3
**Significance:** 2
**Originality:** 2
**Rating:** 4
**Confidence:** 3

**Summary:**

This paper proposes DIFFSSR, a stereo image super-resolution framework that adapts the Differential Transformer to visual tasks. To address attention noise and stereo misalignment, the authors introduce two modules: the Diff Cross Attention Block (DCAB) and the Sliding Stereo Cross-Attention Module (SSCAM). Extensive experiments on standard benchmarks demonstrate that DIFFSSR outperforms prior methods such as SwinFIRSSR and NAFSSR in both accuracy and parameter efficiency.

**Questions:**

1. Since the method is built on the Differential Transformer, the paper would benefit from a brief preliminaries section summarizing its core ideas and formulations. This would help readers better understand the motivation behind the proposed architectural modifications.
2. The paper should clearly state that the effectiveness of the Differential Transformer comes from constructing opposing attention heads and using their difference to highlight relevant features while suppressing background noise. This design motivation is important for understanding how it addresses attention noise. Additionally, Section 3.3 should better explain why SSCAM is effective for handling stereo misalignment.
3. Section 4.3.1 discusses the impact of window sizes 1 to 3, but Table 2 does not include results for sizes 1 and 2. These should be reported for completeness. In addition, Table 4 should include a comparison between window-based multi-head differential attention and standard multi-head differential attention, as this is one of the three core modifications.

If the authors can adequately address the above questions and resolve the concerns outlined in the weaknesses section, I would be willing to reconsider my overall assessment.

**Ethical Concerns:**

["NO or VERY MINOR ethics concerns only"]

**Final Justification:**

The authors have clarified the novelty in the rebuttal, which, while still relatively modest, is acceptable. I hope they will, as promised, include more theoretical motivation in the final version to better highlight the contributions. Therefore, I am willing to raise my rating.

**Limitations:**

yes

**Quality:**

3

**Strengths And Weaknesses:**

Strengths:

1. The paper is well organized and clearly written, making it easy to understand the motivation, proposed method, and the role of each component.

2. This paper presents the first attempt to adapt the Differential Transformer to visual tasks and proposes two modules, DCAB and SSCAM, which effectively reduce attention noise and address stereo misalignment.

3. The method achieves competitive performance on multiple standard benchmarks, with comprehensive comparisons against SOTA baselines, showing consistent improvements in both accuracy and efficiency.


Weaknesses:

1. Although the paper introduces two new modules, their design is relatively incremental. Both DCAB and SSCAM are based on known concepts such as attention refinement and sliding window mechanisms, and may not be considered highly novel from an architectural perspective.

2. While the paper provides high-level motivation for DCAB and SSCAM, it lacks deeper analysis or justification of why these specific designs are effective in addressing attention noise and stereo misalignment. The theoretical or intuitive reasoning behind the effectiveness of the modules could be elaborated further.

3. There are several minor errors and unclear points in the paper. For example, the red, blue, and green boxes in Figure 3(b) are not explicitly explained. Equations (13) and (14) use incorrect input notations, and some subscripts or superscripts are inconsistent with those shown in Figure 3(b). Additionally, the meanings of the superscripts L and R in Section 3.1 should be clearly defined at the beginning of the section.

---

> ### Author Rebuttal · Authors · 2025-07-31
>
> We appreciate the constructive comments on our paper. The raised concerns are addressed as follows.
>
> 1. Although the paper introduces two new modules, their design is relatively incremental. Both DCAB and SSCAM are based on known concepts such as attention refinement and sliding window mechanisms, and may not be considered highly novel from an architectural perspective.
>
>    > A1: a. Our DCAB is the first work to explore the applicability of DIFF Transformer in visual task, specifically optimized for stereo image super-resolution (SISR), please see Sec. 3.2 for details. b. SSCAM solves the performance degradation problem caused by misalignment of horizontal epipolar line in stereo images. This design has never appeared in existing methods and is a specialized optimization for SISR. c. In summary, our DCAB and SSCAM are not simply integrated concepts, but specific improvements proposed for SISR, and experimental results also demonstrate the effectiveness of our method (Figure 4 and Table 1).
>
> 2. While the paper provides high-level motivation for DCAB and SSCAM, it lacks deeper analysis or justification of why these specific designs are effective in addressing attention noise and stereo misalignment. The theoretical or intuitive reasoning behind the effectiveness of the modules could be elaborated further.
>
>    > A2: a. In a traditional Transformer, the attention mechanism distributes focus across all tokens in a sequence using a single softmax function. This behavior over-smoothing the results when the Transformer gives unnecessary weight to distracting tokens. Instead of relying on a single attention map, DIFF Transformer computes two softmax maps: One focuses on relevant tokens; The other represents noise or distractions in the input. The difference between the two maps becomes the final attention score, removing shared distractions while amplifying the signal that matters. This mechanism aligns with the idea of filtering out common-mode noise, allowing the model to efficiently focus on key information.  And Sec. A.0.2 intuitively indicate that DIFF can effectively reduce attention noise compared to self attention. b. As you can see from the principle of SCAM and Figure 2, SCAM can not solve the problem of misalignment of horizontal epipolar line. Our SSCAM addresses this problem by sliding window mechanism and parallax attention. To validate the effectiveness of SSCAM, we simulated misalignment between left and right views by randomly applying vertical shifts of 0-3 pixels to the right view in the Flickr1024 validation dataset. We evaluated only the PSNR of the left view in this setup. The results indicate that our SSCAM incurs negligible performance degradation, while the methods based on SCAM (SwinFIRSSR and NAFSSR) exhibit a significant drop in performance (see Sec. A.0.1).
>
> 3. There are several minor errors and unclear points in the paper. For example, the red, blue, and green boxes in Figure 3(b) are not explicitly explained. Equations (13) and (14) use incorrect input notations, and some subscripts or superscripts are inconsistent with those shown in Figure 3(b). Additionally, the meanings of the superscripts L and R in Section 3.1 should be clearly defined at the beginning of the section.
>
>    > A3: As you suggest, we are going to add instructions and uniform symbolic representations. Red is layer normalization, blue is window-based multi-head differential attention module, green is multilayer perceptron. $F_L$ and $F_R$ in  Figure 3 should be $F^L$ and $F^R$. L and R are left and right images.
>
> 4. Since the method is built on the Differential Transformer, the paper would benefit from a brief preliminaries section summarizing its core ideas and formulations. This would help readers better understand the motivation behind the proposed architectural modifications.
>
>    > A4: As you suggest, we are going to add a brief preliminary to summarize the DIFF Transformer and our research motivations.
>
> 5. The paper should clearly state that the effectiveness of the Differential Transformer comes from constructing opposing attention heads and using their difference to highlight relevant features while suppressing background noise. This design motivation is important for understanding how it addresses attention noise. Additionally, Section 3.3 should better explain why SSCAM is effective for handling stereo misalignment.
>
>    > A5: a. Thank you for your suggestion. As you understand, that is exactly our core research motivation. Excessive fusion of irrelevant information can reduce the quality of image super-resolution, and this negative impact can be mitigated through the use of DIFF Transformer. b. We will add a legend to further illustrate the working principle of SSCAM.
>
> 6. Section 4.3.1 discusses the impact of window sizes 1 to 3, but Table 2 does not include results for sizes 1 and 2. These should be reported for completeness. In addition, Table 4 should include a comparison between window-based multi-head differential attention and standard multi-head differential attention, as this is one of the three core modifications.
>
>    > A6: a. "-" indicates window size=1. The window sizes we designed are all odd values, so there are no experimental results with window size=2. b. The "CUDA out of memory" error on the 3090 when run standard DIFF indicates that it is not suitable for visual tasks. We optimized it to run efficiently on low performance GPUs.

---

> > ### Comment · Reviewer_CHFU · 2025-08-07
> >
> > Thank you for your response and clarification. Most of my concerns have been addressed. I have a few remaining suggestions:
> >
> > 2\. b) While I appreciate the experimental validation provided for SSCAM, I believe its theoretical motivation should be more clearly articulated in the main paper. In contrast to response (a), which offers a solid and intuitive explanation of how the Differential Transformer mitigates attention noise through contrasting attention maps, the justification for SSCAM relies primarily on empirical evidence. It does not sufficiently explain why the sliding window mechanism and parallax attention help address stereo misalignment from a theoretical standpoint. For completeness and clarity, I encourage the authors to include a concise explanation in Section 3.3 on how the sliding window mechanism enables the model to handle vertical shifts in epipolar geometry—rather than leaving this insight implicit or supported solely by experimental results.
> >
> > 6\. a) Regarding the window size, while using odd-sized windows is a common and reasonable design choice in vision models due to symmetry and implementation simplicity, the paper includes a dedicated ablation study on this aspect. Therefore, it would be helpful to include a brief explanation in the main text about why even-sized windows (e.g., size 2) were excluded, or to add corresponding experimental results to ensure the completeness of the analysis.
> >
> > 6\. b) Additionally, I do not find it entirely convincing to conclude that the standard DIFF architecture is "unsuitable for vision tasks" solely because it runs out of memory on an RTX 3090. This appears to be an implementation-level constraint rather than a fundamental limitation of the method itself. The rebuttal's phrasing in this regard feels logically imprecise in terms of causality. A more appropriate explanation would be that the original design incurs significant memory overhead, which motivates the introduction of the more efficient window-based variant. That said, I acknowledge the practical value of your design and find the experimental results to be sound and convincing.

---

> > > ### Author Response · Authors · 2025-08-07
> > >
> > > Dear Reviewer CHFU,
> > >
> > > Thank you very much for your time and expert guidance throughout the rebuttal process. We will incorporate your valuable suggestions into the paper. We sincerely wish you can raise the rating, your support is really important for our work.
> > >
> > > Best wishes, Authors of paper 7904

---

### Official Review · Reviewer_6cnT · 2025-07-04

**Clarity:** 3
**Significance:** 2
**Originality:** 2
**Rating:** 3
**Confidence:** 4

**Summary:**

This paper proposes DIFFSSR, which transfers the Differential Transformer to stereo image super-resolution. It pairs a Diff Cross Attention Block that keeps useful detail and reduces noise with a Sliding Stereo Cross-Attention Module that scans left and right views to fix misalignment. DIFFSSR beats SwinFIRSSR and NAFSSR in both scores and visual quality.

**Questions:**

Please refer to the weaknesses.

**Ethical Concerns:**

["NO or VERY MINOR ethics concerns only"]

**Final Justification:**

While the method shows some empirical gains, its core components are largely built on existing modules, with limited evidence of true methodological novelty. Some experimental comparisons, such as on the Middlebury dataset, may be biased due to differences in training data. Key design changes like replacing the Unfolding operation were not part of the original paper and lack sufficient justification. Overall, my concerns regarding novelty, fairness, and completeness remain unresolved.

**Limitations:**

yes

**Quality:**

2

**Strengths And Weaknesses:**

Strengths:
1.  Visual comparisons demonstrate superior texture preservation and reduced over-smoothing relative to the baselines.

Weaknesses:
1. Limited novelty: DCAB and SSCAM integrate established concepts such as Differential Transformer, windowed self-attention and cross-view attention, without new improvements for stereo tasks.
2. Heavy computation with incomplete efficiency analysis: Even with a small window, SSCAM incurs about 40 % more FLOPs than the variant without the window, which can hinder real-time deployment. Besides, this paper does not report FLOPs for competing methods.
3. Incremental experimental gains: Compared to competing methods with comparable parameter counts, DIFFSSR yields < 0.1 dB improvement on most datasets, indicating minimal practical advantage.
4. Excessive mathematical notation: The manuscript contains 25 equations, many of which are not essential to the core derivation and could be removed.

---

> ### Author Rebuttal · Authors · 2025-07-31
>
> We sincerely thank you for reviewing our paper and providing us valuable feedback. We have addressed your concerns as below.
>
> 1. Limited novelty: DCAB and SSCAM integrate established concepts such as Differential Transformer, windowed self-attention and cross-view attention, without new improvements for stereo tasks.
>
>    > A1: a. Our DCAB is the first work to explore the applicability of DIFF Transformer in visual task, specifically optimized for stereo image super-resolution (SISR), please see Sec. 3.2 for details. b. SSCAM solves the performance degradation problem caused by misalignment of horizontal epipolar line in stereo images. This design has never appeared in existing methods and is a specialized optimization for SISR. c. In summary, our DCAB and SSCAM are not simply integrated concepts, but specific improvements proposed for SISR, and experimental results also demonstrate the effectiveness of our method (Figure 4 and Table 1).
>
> 2. Heavy computation with incomplete efficiency analysis: Even with a small window, SSCAM incurs about 40 % more FLOPs than the variant without the window, which can hinder real-time deployment. Besides, this paper does not report FLOPs for competing methods.
>
>    > A2: a. SSCAM is training free and can be trained in large windows while reasoning in the original window, improving performance without increasing inference time, as shown in Table 7. b. For real-time issues, a mixed window design can be used, such as using a 3x3 window for the first layer DCAL in each DCAB and a 1x1 window for the rest. The performance did not decrease in Flickr1024, while Flops decreased from 187.11 to 153.98. c. We have reconsidered the Unfolding operation and believe that we can use a window shift method similar to swin transformer instead of Unfolding. This way is training free and can reduce duplicate calculations and be more efficient, please see the Table. Although other operations can be used instead of the Unfolding operation, the core idea of our method remains unchanged. d. As reviewer JZxY pointed out, misalignment is a very common issue. And we only randomly shifts 0-3 pixels, resulting in a significant decrease in the performance of existing SCAM, as shown in Table 5. And our method maintains considerable performance, indicating that SSCAM is very necessary. e. We are going to add comparison of FLOPs with mainstream methods such as SwinFIRSSR and SCGLANet.
>    >
>    > |           Window size            | PSNR  | Flops  |
>    > | :------------------------------: | :---: | :----: |
>    > |  DIFFSSR (win=1, Unfolding op)   | 24.35 | 137.42 |
>    > | DIFFSSR (win=3-1, Unfolding op)  | 24.39 | 153.98 |
>    > |  DIFFSSR (win=3, Unfolding op)   | 24.39 | 184.11 |
>    > | DIFFSSR (win=1, window shift op) | 24.36 | 137.42 |
>    > | DIFFSSR (win=2, window shift op) | 24.40 | 143.63 |
>    > |            SwinFIRSSR            | 24.29 | 147.39 |
>    > |             SCGLANet             | 24.30 | 138.83 |
>    >
>    >
>
> 3. Incremental experimental gains: Compared to competing methods with comparable parameter counts, DIFFSSR yields < 0.1 dB improvement on most datasets, indicating minimal practical advantage.
>
>    > A3: DIFFSSR achieves a PSNR improvement of 0.23 dB over NAFSSR-L on the Flickr1024 dataset (vs. 0.13 dB for SCGLANet and 0.12 dB for SwinFIRSSR), and 0.44 dB on Middlebury (vs. -0.16 dB for SCGLANet and 0.24 dB for SwinFIRSSR). These improvements are considered significant in the super-resolution field, especially the visual improvement is even more significant, as shown in Figure 4.
>
> 4. Excessive mathematical notation: The manuscript contains 25 equations, many of which are not essential to the core derivation and could be removed.
>
>    > A4: As you suggest, we are going to remove equations that do not affect understanding of implementation details.

---

> > ### Comment · Reviewer_6cnT · 2025-08-07
> >
> > Thanks for the response. However, I still have some major concerns.
> >
> > Simply applying existing modules to a new task does not, in itself, constitute a methodological innovation. Additionally, the authors attempt to demonstrate the effectiveness of their method using the performance gain on the Middlebury dataset. However, to the best of my knowledge, SCGLANet was trained only on Flickr1024, while DIFFSSR was trained on both Flickr1024 and Middlebury. This raises concerns about the validity of the comparison, as the improvement on Middlebury may simply result from additional training data or distribution similarity. Furthermore, in the bottom row of Figure 4, the visual results do not clearly indicate that DIFFSSR outperforms SwinFIRSSR.
> >
> > We also appreciate the clarification regarding the unfolding operation, but replacing such a core component should be supported with more explanation or evidence to show that the method’s main idea and contribution remain intact. While the newly proposed designs are indeed more efficient, they were not presented in the original paper. This raises further concerns about whether the method’s best configuration was fully explored, and whether the reported results truly reflect the optimal performance.
> >
> > Therefore, I tend to keep the original score at this stage.

---

> > > ### Author Response · Authors · 2025-08-09
> > >
> > > Dear Reviewer 6cnT,
> > >
> > > If there are still any remaining concerns, please kindly let us know. We will spare no effort to address them thoroughly. We sincerely appreciate your valuable comments, constructive suggestions, and the time and effort you have devoted to reviewing our work. Your guidance is truly invaluable to us.
> > >
> > > Best wishes, Authors of paper 7904

---

> ### Author Response · Authors · 2025-08-07
>
> Thank you very much for your feedback, and we apologize that our previous response did not fully address your concerns. Below we address them one by one.
>
> 1. Simply applying existing modules to a new task does not, in itself, constitute a methodological innovation.
>
>    > We are not simply applying existing DIFF Transformer to visual tasks, as directly applying it to stereo image super-resolution (SISR) task would result in significant performance degradation, as shown in Figure 1. In order to enable it to play a better role in the visual tasks, our DCAB has made a series of improvements, and the detailed improvement process is presented in Sec 3.2. In addition, our DCAB is also a brand new architecture that increases the opportunity for information exchange between left and right views compared to SwinFIRSSR.
>
> 2. Additionally, the authors attempt to demonstrate the effectiveness of their method using the performance gain on the Middlebury dataset. However, to the best of my knowledge, SCGLANet was trained only on Flickr1024, while DIFFSSR was trained on both Flickr1024 and Middlebury. This raises concerns about the validity of the comparison, as the improvement on Middlebury may simply result from additional training data or distribution similarity.
>
>    > According to the NTIRE 2023 Challenge on Stereo Image Super-Resolution [1] results, SwinFIRSSR won the championship in Track 2. Furthermore, the NTIRE 2024 Challenge on Stereo Image Super-Resolution [2] results demonstrate that the tiny version of SwinFIRSSR achieved dual championships in both Track 1 and Track 2. In contrast, SCGLANet ranked 3rd in Track 2 of NTIRE 2023 and the light version of SCGLANet (SOAN) ranked  4th in Track1 and Track2 of NTIRE 2023. **They are all trained on Flickr1024.** Notably, our proposed DIFFSSR surpasses SwinFIRSSR across all four benchmark datasets, which provides compelling evidence for the effectiveness of our methodology. Of course, we will train our DIFFSRS only on Flickr1024, if the training can be completed before August 8th, we will report the corresponding results.
>    >
>    > | **NTIRE 2023** | **Track2** |        |        | **NTIRE 2024** |     **Track1**      |         | **NTIRE 2024** |     **Track2**      |         |
>    > | :------------: | :--------: | :----: | ------ | :------------: | :-----------------: | :-----: | :------------: | :-----------------: | :-----: |
>    > |    **Rank**    | **Method** | Score  | LPIPS  |    **Rank**    |     **Method**      |  PSNR   |    **Rank**    |     **Method**      |  PSNR   |
>    > |       1        | SwinFIRSSR | 0.8622 | 0.1386 |       1        |     SwinFIRSSR      | 23.6503 |       1        |     SwinFIRSSR      | 21.8724 |
>    > |       3        |  SCGLANet  | 0.8496 | 0.1493 |       4        | SOAN(base SCGLANet) | 23.5941 |       4        | SOAN(base SCGLANet) | 21.6691 |
>    >
>    > [1]  NTIRE 2023 challenge on stereo image super-resolution: Methods and results, CVPRW2023
>    >
>    > [2]  NTIRE 2024 challenge on stereo image super-resolution: Methods and results, CVPRW2024
>    >
>    >
>    >
>    > In addition, we also compared SCGLANet on a real data set. As StereoWeb20 is a real-world benchmark dataset lacking ground truth, we evaluated the models using no-reference image quality assessment metrics, such as NIQE, MANIQA, MUSIQ and CLIPIQA. Regarding model training, SCGLANet-GAN underwent 400,000 iterations with a batch size of 3 on 8 GPUs. In contrast, our DIFFSSR-GAN was trained for 60,000 iterations with a batch size of 2 on 4 GPUs. Importantly, DIFFSSR-GAN surpasses the performance of SCGLANet-GAN despite utilizing only **15%** of its training iterations.
>    >
>    > |            Datasets             | Metrics  | NAFSSR [6] | SCGLANet-GAN [32] | DIFFSSR-GAN |
>    > | :-----------------------------: | :------: | :--------: | :---------------: | :---------: |
>    > | StereoWeb20(real-world dataset) |  NIQE↓   |   5.7363   |      4.4174       | **4.2248**  |
>    > |                                 | MANIQA↑  |   0.4648   |      0.5761       | **0.6117**  |
>    > |                                 |  MUSIQ↑  |   46.78    |       62.18       |  **63.31**  |
>    > |                                 | CLIPIQA↑ |   0.5144   |      0.6331       | **0.6623**  |
>
> 3. Furthermore, in the bottom row of Figure 4, the visual results do not clearly indicate that DIFFSSR outperforms SwinFIRSSR.
>
>    > In the bottom row of Figure 4, our results have clearer textures than SwinFIRSSR, and the results recovered by SwinFIRSSR have obvious blurring issues. We will add saliency markers in Figure 4.

---

> > ### Author Response · Authors · 2025-08-07
> >
> > 4. We also appreciate the clarification regarding the unfolding operation, but replacing such a core component should be supported with more explanation or evidence to show that the method’s main idea and contribution remain intact. While the newly proposed designs are indeed more efficient, they were not presented in the original paper. This raises further concerns about whether the method’s best configuration was fully explored, and whether the reported results truly reflect the optimal performance.
> >
> >    > We sincerely apologize for the confusion caused here. Fundamentally, the Unfolding op and the window shift op are computationally equivalent, differing only in their implementation approaches. This is similar to convolution operations. There are multiple implementations in the PyTorch framework, such as Winograd, im2col, and FFT. This is why when win=1, the Flops of both are the same. In win=2, the step=win of Unfolding op is almost the same as the Flops of window shift op.
> >    >
> >    > |               Window size               | PSNR  | Flops  |
> >    > | :-------------------------------------: | :---: | :----: |
> >    > |      DIFFSSR (win=1, Unfolding op)      | 24.35 | 137.42 |
> >    > |    DIFFSSR (win=1, window shift op)     | 24.36 | 137.42 |
> >    > |    DIFFSSR (win=2, window shift op)     | 24.40 | 143.63 |
> >    > | DIFFSSR (win=2, Unfolding op, step=win) | 24.40 | 144.41 |
> >    >
> >    > Unfolding op and window shift op essentially perform blocking (or patching) operations on images, and the stereo cross attention after blocking is implemented consistently (as shown in the code).
> >    >
> >    > ```
> >    > # Unfolding op
> >    >
> >    > win = self.win # self.win=2
> >    > Q_l = self.l_proj1(x_l)
> >    > Q_r_T = self.r_proj1(x_r)
> >    > nn_Unfold = nn.Unfold(kernel_size=(win, w), padding=(win // 2, 0), stride=2)
> >    > Q_l = nn_Unfold(Q_l)
> >    > Q_r_T = nn_Unfold(Q_r_T)
> >    > Q_l = rearrange(Q_l, 'b (ch owh oww) nw -> (b nw) (owh oww) ch', ch=c, owh=win, oww=w).contiguous()
> >    > Q_r_T = rearrange(Q_r_T, 'b (ch owh oww) nw -> (b nw) (owh oww) ch', ch=c, owh=win, oww=w).contiguous()
> >    > Q_r_T = Q_r_T.permute(0, 2, 1).contiguous()
> >    >
> >    > ############# stereo cross attention #######################################
> >    > attention = torch.matmul(Q_l, Q_r_T) * self.scale
> >    > V_l = self.l_proj2(x_l)
> >    > V_r = self.r_proj2(x_r)
> >    > V_l = nn_Unfold(V_l)
> >    > V_r = nn_Unfold(V_r)
> >    > V_l = rearrange(V_l, 'b (ch owh oww) nw -> (b nw) (owh oww) ch', ch=c, owh=win, oww=w).contiguous()
> >    > V_r = rearrange(V_r, 'b (ch owh oww) nw -> (b nw) (owh oww) ch', ch=c, owh=win, oww=w).contiguous()
> >    > F_r2l = torch.matmul(torch.softmax(attention, dim=-1), V_r)  # B, L, c
> >    > F_l2r = torch.matmul(torch.softmax(attention.permute(0, 2, 1).contiguous(), dim=-1), V_l)
> >    > #################################################################
> >    >
> >    > fold = torch.nn.Fold(output_size=(h, w), kernel_size=(win, w), padding=(win // 2, 0), stride=2)
> >    > F_r2l = rearrange(F_r2l, '(b nw) (owh oww) ch -> b (ch owh oww) nw', b=b//2, ch=c, owh=win, oww=w).contiguous()
> >    > F_l2r = rearrange(F_l2r, '(b nw) (owh oww) ch -> b (ch owh oww) nw', b=b//2, ch=c, owh=win, oww=w).contiguous()
> >    > F_r2l = fold(F_r2l)
> >    > F_l2r = fold(F_l2r)
> >    > out = torch.cat([x_l + F_r2l * self.beta, x_r + F_l2r * self.gamma], 0)
> >    > ```
> >    >
> >    > ```
> >    > # window shift op
> >    >
> >    > win_size = self.win # self.win=2
> >    > Q_l = self.l_proj1(x_l).permute(0, 2, 3, 1).contiguous()
> >    > Q_r = self.r_proj1(x_r).permute(0, 2, 3, 1).contiguous()
> >    > Q_l_win = window_partition(Q_l, (win_size, W))
> >    > Q_l_win = Q_l_win.view(-1, win_size * W, C)
> >    > Q_r_win = window_partition(Q_r, (win_size, W))
> >    > Q_r_win = Q_r_win.view(-1, win_size * W, C)
> >    > Q_r_win = Q_r_win.permute(0, 2, 1).contiguous()
> >    >
> >    > ############# stereo cross attention #######################################
> >    > attention = torch.matmul(Q_l_win, Q_r_win) * self.scale
> >    > V_l = self.l_proj2(x_l).permute(0, 2, 3, 1).contiguous()
> >    > V_r = self.r_proj2(x_r).permute(0, 2, 3, 1).contiguous()
> >    > V_l_win = window_partition(V_l, (win_size, W))
> >    > V_l_win = V_l_win.view(-1, win_size * W, C)
> >    > V_r_win = window_partition(V_r, (win_size, W))
> >    > V_r_win = V_r_win.view(-1, win_size * W, C)
> >    > F_r2l = torch.matmul(torch.softmax(attention, dim=-1), V_r_win)
> >    > F_l2r = torch.matmul(torch.softmax(attention.permute(0, 2, 1).contiguous(), dim=-1), V_l_win)
> >    > #################################################################
> >    >
> >    > F_r2l = F_r2l.view(-1, win_size, W, C)
> >    > F_l2r = F_l2r.view(-1, win_size, W, C)
> >    > F_r2l = window_reverse(F_r2l, (win_size, W), H, W).permute(0, 3, 1, 2)
> >    > F_l2r = window_reverse(F_l2r, (win_size, W), H, W).permute(0, 3, 1, 2)
> >    > out = torch.cat([x_l + F_r2l * self.beta, x_r + F_l2r * self.gamma], 0)
> >    > ```

---

> > > ### Author Response · Authors · 2025-08-07
> > >
> > > 5. Therefore, I tend to keep the original score at this stage.
> > >
> > >    > Thank you very much for your time. We truly appreciate your valuable feedback throughout the review process. If you have any remaining concerns or suggestions, we would be more than happy to address them. Please don’t hesitate to share your thoughts with us, and we would be glad to further discuss them.

---

> ### Author Response · Authors · 2025-08-07
>
> We updated the SCGLANet [3] performance that trained on Flickr1024 and Middlebury and hope it addresses your concerns. And NAFSSR-L, SwinFIRSSR, SCGLANet and DIFFSSR are all trained on Flickr1024 and Middlebury. Our DIFFSSR surpasses NAFSSR-L, SwinFIRSSR, and SCGLANet, achieving SOTA performance in stereo image super-resolution across most datasets and scales.
>
> | Scale | **Method** | Flickr1024 | KITTI2012 | KITTI2015 | Middlebury | Average |
> | :---: | :--------: | :--------: | :-------: | :-------: | :--------: | :-----: |
> |  X2   |  NAFSSR-L  |   29.68    |   31.60   |   31.25   |   35.88    |  32.10  |
> |  X2   | SwinFIRSSR |   30.14    |   31.79   |   31.45   |   36.52    |  32.48  |
> |  X2   |  SCGLANet  |   29.97    |   31.71   |   31.32   |   36.33    |  32.33  |
> |  X2   |  DIFFSSR   |   30.27    |   31.84   |   31.47   |   36.65    |  32.56  |
> |  X4   |  NAFSSR-L  |   24.17    |   27.12   |   26.96   |   30.20    |  27.11  |
> |  X4   | SwinFIRSSR |   24.29    |   27.16   |   26.89   |   30.44    |  27.20  |
> |  X4   |  SCGLANet  |   24.39    |   27.20   |   27.01   |   30.38    |  27.25  |
> |  X4   |  DIFFSSR   |   24.40    |   27.25   |   26.98   |   30.64    |  27.32  |
>
> [3] Zhou Y, Xue Y, Bi J, et al. Towards real world stereo image super-resolution via hybrid degradation model and discriminator for implied stereo image information[J]. Expert Systems with Applications, 2024, 255: 124457.

---

### Note · Authors · 2025-08-12

Dear PCs, ACs, and all reviewers,

We sincerely thank you for your time, effort, and thoughtful engagement during this process. The feedback we have received has been invaluable, enabling us to substantially strengthen our work both technically and in terms of presentation. We have also worked diligently to address every concern with new experiments, deeper analysis, and clearer explanations.

Our work, DIFFSSR, is the first to adapt the DIFF Transformer to visual tasks, introducing two key modules—**Diff Cross Attention Block (DCAB)** and **Sliding Stereo Cross-Attention Module (SSCAM)**—specifically designed for stereo image super-resolution (SISR). These are not direct applications of existing ideas, but task-driven innovations:

* **DCAB** reformulates DIFF Transformer to enable efficient cross-view information exchange and robust suppression of attention noise, avoiding the performance degradation that occurs when directly applying DIFF to SISR, as shown in Figure 1.
* **SSCAM** is the first stereo attention design that explicitly mitigates misalignment of horizontal epipolar lines in stereo images using sliding windows with parallax-aware attention—crucial for real-world stereo data (see Table 5 and our responses to Reviewer JZxY).

While PSNR gains may appear numerically small on well-aligned benchmarks (0.46dB improvement over NAFSSR-L vs. 0.23dB for SCGLANet and 0.38dB for SwinFIRSSR), our extensive robustness analysis shows that under realistic misalignments (0–24 pixels), DIFFSSR maintains performance where prior SOTA methods degrade substantially. On the real-world StereoWeb20 dataset, DIFFSSR-GAN surpasses SCGLANet-GAN despite using only 46% of its training iterations, demonstrating both practical efficiency and superior generalization. Statistical tests confirm that our improvements over SwinFIRSSR are highly significant (p-values ≤ 1e-5) and consistent across >93% of images.

We deeply respect differing academic perspectives, but we believe the combination of novel, task-driven architectural improvements, demonstrated robustness under challenging real-world conditions, and consistent gains over competitive baselines makes DIFFSSR a meaningful contribution to the community. And we have also addressed all concerns raised in good faith and with the utmost diligence. If we solved all your concerns, we humbly and respectfully hope that you can consider supporting our work.

With sincere appreciation and respect,

Authors of Submission 7904

---

### Decision · Program_Chairs · 2025-09-17

**Decision:**

Accept (poster)

**Comment:**

This paper proposes an effective DIFFSSR that includes DCAB and SSCAM modules for stereo image super-resolution. Experimental results demonstrate that the effect of the proposed DIFFSSR.

The major concerns of this paper include the limited novelty and marginal performance improvement. Based on the provided rebuttal, Reviewer 6cnT still pointed out that the novelty of the paper is limited. Other reviewers are satisfied with authors' response. However, their recommendations are "Borderline accept".

If this manuscript is accepted, the authors should clarify the novelty.